

**Impacts of urbanization on air quality and related health risks in**
**a city with complex terrain**
Chenchao Zhan [a], Min Xie [a,*], Hua Lu [b], Bojun Liu [c], Zheng Wu [b], Tijian Wang [a], Bingliang Zhuang
[a], Mengmeng Li [a], Shu Li [a]
[a] School of Atmospheric Sciences, Nanjing University, Nanjing 210023, China
[b] Chongqing Institute of Meteorological Sciences, Chongqing 401147, China
[c] Chongqing Meteorological Observatory, Chongqing 401147, China

8 --------------------------------------------------------------------

[*] Corresponding author. minxie@nju.edu.cn (M. Xie)
**Abstract:** Urbanization affects air pollutants by urban expansion and emission growth, and thereby
inevitably changes health risks of air pollutants. However, the health risks related to urbanization
are rarely estimated, especially for cities with complex terrain. In this study, a highly urbanized city
with severe air pollution and complex terrain (Chengdu) is selected to explore this issue. The effects
of urban expansion are further compared with emission growth as air quality management is mainly
to regulate emissions. Air pollution in Chengdu is mainly caused by $PM_{2.5}$ and $O_3$. $PM_{2.5}$ pollution
tends to appear in cold months (November to February) due to the secondary circulation forced by
complex terrain and the frequent temperature inversion, while $O_3$ pollution is likely to occur in warm
months (April to August) because of high temperature and strong sunlight dominated by high-
pressure systems. From 2015 to 2021, the annual total premature mortalities from all non-accidental
causes (ANAC) attributed to $PM_{2.5}$ and $O_3$ exposure are 9386 and 7743, respectively. Based on the
characteristics of $PM_{2.5}$ and $O_3$, six numerical experiments are conducted to investigate the impacts
of urban expansion and emission growth on health risks. The results show that urban expansion
causes an increase in air temperature and the boundary layer height, which is conducive to the
diffusion of $PM_{2.5}$. Thus, surface $PM_{2.5}$ concentrations decrease by 11.7 $\mu g\ m^{-3}$ in January. However,
the MDA8 $O_3$ concentrations increase by 10.6 $\mu g\ m^{-3}$ in July due to the stronger photochemical
production and better vertical mixing during the daytime. Correspondingly, the total premature
mortalities from ANAC attributed to $PM_{2.5}$ exposure decrease by 182 (6.9%) in January, but those
attributed to $O_3$ exposure increase by 203 (9.5%) in July. As for the effects of emission growth,





PM$_{2.5}$ and MDA8 O$_3$ concentrations can increase by 26.6 µg m$^{-3}$ and 4.8 µg m$^{-3}$ when anthropogenic
emissions are taken into account. The total premature mortalities from ANAC attributed to PM$_{2.5}$
and O$_3$ exposure then increase by 424 (16.0%) and 87 (4.1%), respectively. The effect of urban
expansion on health risks of PM$_{2.5}$ is about half that of anthropogenic emissions. Whereas the effect
of urban expansion on health risks of O$_3$ can be 2.3 times that of anthropogenic emissions. This
reminds us that the development of cities is also important for the urban air quality apart from the
emissions reduction.

**Key Words:** urbanization; land use; anthropogenic emissions; air quality; health risk;

**1 Introduction**
Air pollutants are substances that damage humans, plants and animals drastically when present
in the atmosphere in sufficient concentration (Baklanov et al., 2016; Kinney, 2018; Pautasso et al.,
2010). The most common air pollutants found in air are ozone (O$_3$), fine particulate matter (PM$_{2.5}$,
particulate matter with an aerodynamic diameter of 2.5 µm or less), sulfur dioxide (SO$_2$) and
nitrogen oxides (NO$_x$, which is NO + NO$_2$). These air pollutants threaten human health in many
parts of the world, evoking a series of health risks including cardiovascular diseases, respiratory
diseases and chronic obstructive pulmonary disease (Brauer et al., 2016; Lelieveld et al., 2013;
Manisalidis et al., 2020). According to the World Health Organization (WHO), exposure to ambient
air pollutants results in about 4.2 million premature deaths globally per year
(https://www.who.int/health-topics/air-pollution#tab=tab_2).
Most of those premature deaths occur in urban areas as urban areas currently host more than
50% of the population (over 3.5 billion people) and this number is projected to increase to 70% by
2050 (UNDESA, 2018). What's more, urban areas are centers of resource utilization and are a major
contributor to air pollutant and greenhouse gas emissions. Urbanization, along with socioeconomic
development, also leads to increase in anthropogenic emissions. Air pollutants that originate from
anthropogenic sources can sometimes accumulate and degrade urban air quality, which leaves urban
dwellers vulnerable to air pollution (Holman et al., 2015; Lin and Zhu, 2018). Excessive emissions
are considered to be the root cause of poor air quality in urban areas, and thereby efforts have been
made to reduce anthropogenic emissions to achieve the goal of managing urban air pollution.



In addition to emissions, urban air quality is also closely related to meteorology (Qian et al.,
2017). Under calm conditions, local circulations induced by the thermal contrast of the topography,
such as mountain-valley breezes and sea-land breezes, are likely to form and play an important role
in urban environment (Crosman and Horel, 2010; Zhan and Xie, 2022). Because of historical,
political and economic reasons, about 12% of the population (over 720 million people) resides in
mountainous areas where air pollution is usually more severe than flat locations since mountainous
terrain strongly alters the boundary layer structure, resulting in much more complicated diffusion
conditions (Chow et al., 2013). Many notably pollution episodes appear in valley bottoms, along
mountain slopes and in mountain basins. These include examples like Mexico City (Molina et al.,
2010), Hong Kong (Guo et al., 2013), the Seoul (Ryu et al., 2013), the Salt Lake Valley (Baasandorj
et al., 2017), the Colorado Front Range (Bahreini et al., 2018), the Alps (Karl et al., 2019) and the
Taiwan Island (Lee et al., 2019). Although the principles behind these examples apply to
mountainous areas around the world, the phenomenon being described depends on the particular
region (Whiteman, 2000; Oke et al., 2017). And a common principle is that diurnal wind systems
driven by mountainous terrain can recirculate urban air pollutants and worsen air quality.
The world has been undergoing urbanization since the industrial revolution in the 19th century
(Seto et al., 2012), which directly leads to changes in land use via urban expansion. Natural surfaces
are replaced by impervious surfaces, then land surface physical properties (e.g., albedo, thermal
inertia and roughness) and processes (e.g., the exchange of water, momentum and energy) are
modified, hence altering the urban meteorology and air quality. This has been widely investigated
using numerical models. For example, Liao et al. (2015) reported that urban expansion can cause
an increase in 2-m temperature by 0.9–2.3 °C, a decrease in 10-m wind speed by 0.6–1.2 m s$^{-1}$ and
an increase in planetary boundary layer height by 100–425 m in the Yangtze River Delta. These
changes in meteorology further reduce surface $PM_{10}$ concentrations by 15.3–57.6 μg m$^{-3}$ but
increase $O_3$ concentrations by 1.7–8.3 ppbv. Changes in concentrations of air pollutants inevitably
affect their health risks. However, the health risks related to urbanization are rarely estimated,
especially for those cities with complex terrain. This is of great concern to policymakers and can
inspire future air quality control strategies in mountainous areas.
Chengdu (104.01°E, 30.70°N) is the largest city in western China, occupying an area of 12,390
square kilometers with a population of more than 20 million. Chengdu has the most complex terrain



in the world. Located in the west of the Sichuan Basin, this city is surrounded by the Tibetan Plateau
to the west, the Wu Mountains to the east, the Yunnan-Guizhou Plateau to the south and the Daba
Mountains to the north (Figure 1a). In addition, the urbanization of Chengdu has developed rapidly
over the past few decades (Dai et al., 2021). The fast urbanization process is generally accompanied
by a surge in urban construction lands and the loss of cropland (Figure 1c). Luo et al. (2021) reported
that Chengdu's urban area has increased by four times from 1996 to 2016. Due to substantial
emissions from human activities (Figure 1b) and poor atmospheric diffusion capacity caused by
complex terrain, Chengdu is one of the most polluted cities in China and has suffered from severe
$PM_{2.5}$ and $O_3$ pollution in recent years (Shu et al., 2021; Yang et al., 2020; Zhan et al., 2019).
Complex terrain, rapid urbanization and severe air pollution make Chengdu an ideal place to study
the impact of urbanization on health risks of air pollutants in mountainous areas.

In this study, we systematically evaluate the impacts of urbanization on air quality and the

corresponding health risks in Chengdu. We also compare the impacts of urban expansion with
emission growth. First, the basic characteristics of air pollutants in Chengdu from 2015 to 2021 are
illustrated. Then, the impacts of urbanization on air pollutants are investigated by using the WRF-
Chem model. Finally, premature mortalities attributed to changes in air pollutants are estimated by
using the standard damage function. The rest of this paper is organized as follows. Section 2
introduces the observation data, the model configurations and the experimental designs. Section 3
shows the main results and discussions. The conclusions are given in Section 4.



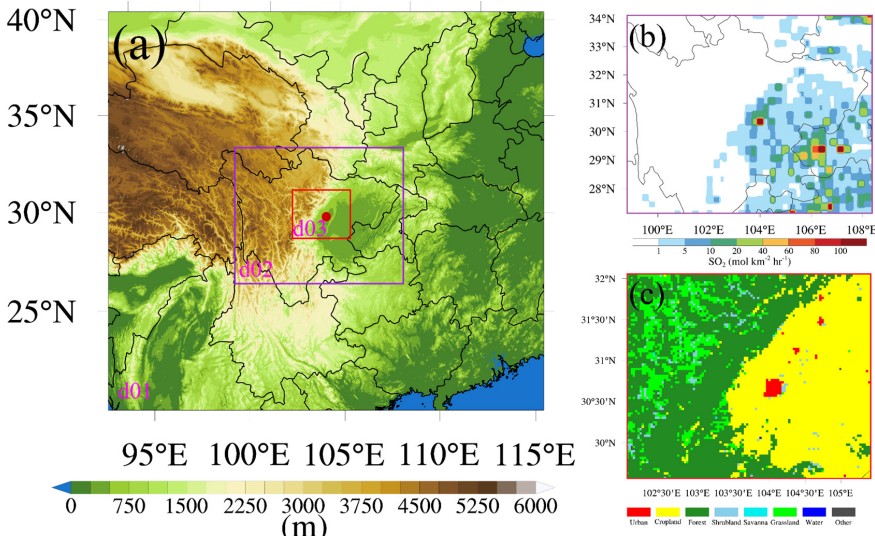

**Figure 1. Three nested WRF-Chem domains, including (a) domain 1 with terrain heights, (b) domain 2 with SO$_2$ emissions and (c) domain 3 with land cover maps. The red dot in (a) shows the location of Chengdu.**

**2 Materials and methods**

**2.1 Air pollutants and meteorological data**

Air pollutants, including PM$_{2.5}$, PM$_{10}$, O$_3$, NO$_2$, SO$_2$ and CO, are monitored by the National Environmental Monitoring Center (NEMC) of China. These data are hourly issued on the national urban air quality real-time publishing platform (http://106.37.208.233:20035/). This nationwide observation network consists of more than 2300 stations distributed over 450 cities in China.

Surface meteorological data, including 2-m air temperature (T$_2$), relative humidity (RH) and 10-m wind speed (WS$_{10}$) and direction (WD$_{10}$), are taken from the website of the University of Wyoming (http://weather.uwyo.edu/surface/). To verify upper-air fields, the sounding observations at Wenjiang (103.87°E, 30.75°N) in Chengdu are also acquired from this website. These sounding data contain temperature, relative humidity and wind speed etc. at different pressure layers with a time resolution of 12 h (00:00 and 12:00 UTC), and are often plotted on a Skew-T diagram (https://www.ncl.ucar.edu/Applications/skewt.shtml#ex2).

In this study, the data quality control are performed as follows. First, the data indicated as



missing are set as invalid. Second, the urban values are calculated by averaging observations at all
monitoring sites in Chengdu. Third, the daily maximum 8 h average (MDA8) $O_3$ concentrations are
calculated only on days with more than 18 h of $O_3$ concentrations measurements.
**2.2 WRF-Chem model and experimental designs**
WRF-Chem is the Weather Research and Forecasting (WRF) model coupled with Chemistry,
in which meteorological and chemical variables use the same coordinates, transport schemes and
physics schemes in space and time (Grell et al., 2005). WRF-Chem version 3.9.1 is employed in
this study. As shown in Figure 1a, three nested domains are used with the grid spacing of 27, 9 and
3 km, respectively. 32 sigma levels are extending from the surface to 100 hPa in the vertical direction
with 12 levels located below 2 km to resolve the boundary layer processes. The MODIS-based land
use data set as default in WRF are selected. The domains and main options for physical and chemical
parameterization schemes are listed in Table 1. The National Centers for Environmental Prediction
(NCEP) global final analysis data with a horizontal resolution of $1° \times 1°$ at 6 h time intervals are
adopted as the initial and boundary conditions. Anthropogenic emissions are provided by the Multi-
resolution Emission Inventory for China (MEIC) in 2017 at $0.25° \times 0.25°$ resolution. Biogenic
emissions are calculated online using the Guenther scheme (Guenther et al., 2006).

**Table 1.** The domains and main options for WRF-Chem.

| Items | Contents |
|---|---|
| Domains (x, y) | (94, 86), (109, 88), (112, 94) |
| Grid spacing (km) | 27, 9, 3 |
| Center | (104°E, 31°N) |
| Time step (s) | 90 |
| Microphysics | Purdue Lin scheme (Chen and Sun, 2002) |
| Longwave radiation | RRTM scheme (Mlawer et al., 1997) |
| Shortwave radiation | Goddard shortwave scheme (Matsui et al., 2018) |
| Surface layer | Monin-Obukhov scheme (Janjic, 1994) |
| Land-surface layer | Unified Noah land-surface model (Tewari et al., 2014) |
| Planetary boundary layer | Mellor-Yamada-Janjic TKE scheme (Janjic, 1994) |





| | |
|---|---|
| Cumulus parameterization | Grell 3D ensemble scheme (Grell and Devenyi, 2002) |
| Gas-phase chemistry | RADM2 (Stockwell et al., 1990) |
| Photolysis scheme | Fast-J photolysis (Fast et al., 2006) |
| Aerosol module | MADE/SORGAM (Schell et al., 2001) |


To investigate the impacts of urban expansion and anthropogenic emissions, six numerical
experiments are designed (Table 2). The year of the numerical simulations is 2017 since the
anthropogenic emissions are currently updated to 2017. Moreover, January is representative of the
cold months with frequent $PM_{2.5}$ pollution, while July represents the hot season with frequent $O_3$
pollution (Section 3.1). Jan_Base simulation is a baseline simulation using the MODIS land use and
the MEIC anthropogenic emission inventory over all three domains. $SO_2$ emissions in domain 2 and
land cover maps in domain 3 are illustrated in Figure 1b and 1c. Jan_noCD is a sensitivity simulation,
in which the urban land surface of Chengdu is replaced by cropland to examine the impacts of urban
expansion. Jan_noEmi is also a sensitivity simulation, in which the anthropogenic emissions in
Chengdu are shut down to identify the impacts of anthropogenic emissions. The above three
numerical experiments used the same configurations (Table 1) running from 00:00 UTC December
28, 2016 to 00:00 UTC February 1, 2017 with the first 96 h as spin-up time. July_Base, July_noCD
and July_noEmi are the same as Jan_Base, Jan_noCD and Jan_noEmi, but run from 00:00 UTC
June 27 to 00:00 UTC August 1, 2017 with the first 96 h as spin-up time.

**Table 2.** Six numerical experiments are conducted in this study.

| Scenarios | Description |
|---|---|
| Jan_Base | Baseline simulation in January |
| Jan_noCD | Replacing urban land use of Chengdu with cropland in January |
| Jan_noEmi | Shutting down anthropogenic emissions in Chengdu in January |
| July_Base | Baseline simulation in July |
| July_noCD | Replacing urban land use of Chengdu with cropland in July |
| July_noEmi | Shutting down anthropogenic emissions in Chengdu in July |






**2.3 Health risks estimation**
Daily premature mortalities attributed to $PM_{2.5}$ and $O_3$ exposure from all non-accidental causes
(ANAC), cardiovascular diseases (CVD), respiratory diseases (RD) and chronic obstructive
pulmonary diseases (COPD) are estimated using the standard damage function (Anenberg et al.,
2010; Zhan et al., 2021):
$$\Delta M = y_0 \left(\frac{RR-1}{RR}\right) \text{Pop}, \tag{4}$$
where $\Delta M$ is the daily premature mortality, $y_0$ is the daily baseline mortality rate, $RR$ is the relative
risk, $(RR-1)/RR$ is the attributable fraction, and Pop is the exposed population. $RR$ is calculated as
follows:
$$RR = \exp(\beta(C - C_0)), \tag{5}$$
where $\beta$ is the concentration-response function, which represents the percentage increase in health
effect per 1 µg m$^{-3}$ $PM_{2.5}$ and MDA8 $O_3$ increment. C is the exposure concentration, and $C_0$ is the
threshold concentration.
In this study, $C_0$ for $PM_{2.5}$ is 10 µg m$^{-3}$ (Song et al., 2015), and for MDA8 $O_3$ is 75.2 µg m$^{-3}$
(Liu et al., 2018). The $\beta$ and $y_0$ values for ANAC, CVD, RD and COPD are summarized in Table 3
(Chen et al., 2017; Yin et al., 2017). The populations of Chengdu provided by the National Bureau
of Statistics of China are 16.853 million, 18.582 million, 19.188 million, 19.183 million, 20.409
million, 20.947 million and 20.938 million from 2015 to 2021.

**Table 3.** Daily $\beta$ and $y_0$ values for ANAC, CVD, RD and COPD. This table is cited from Wang et
al. (2021).

| Disease | $\beta$ for $PM_{2.5}$ (%) | $\beta$ for MDA8 $O_3$ (%) | $y_0$ |
|---------|---------|---------|---------|
| ANAC | 0.22 | 0.24 | $1.687 \times 10^{-5}$ |
| CVD | 0.27 | 0.27 | $3.880 \times 10^{-6}$ |
| RD | 0.29 | 0.18 | $1.841 \times 10^{-6}$ |
| COPD | 0.38 | 0.20 | $1.623 \times 10^{-6}$ |


**3 Results and discussions**
**3.1 $PM_{2.5}$ and $O_3$ pollution in Chengdu**




PM$_{2.5}$ and O$_3$ are two crucial air pollutants that account for air pollution. The Chinese ambient

air quality standards for PM$_{2.5}$ and MDA8 O$_3$ are 75 µg m$^{-3}$ and 160 µg m$^{-3}$, respectively. As shown
in Figure 2, Chengdu is suffering from severe PM$_{2.5}$ and O$_3$ pollution in recent years. There are 97,
101, 68, 53, 33, 43 and 37 PM$_{2.5}$ pollution episodes, and 61, 48, 42, 40, 26, 50 and 27 O$_3$ pollution
episodes in Chengdu from 2015 to 2021. The annual average concentrations of PM$_{2.5}$ are 60.7, 59.9,
52.6, 47.2, 40.6, 40.8 and 40.1 µg m$^{-3}$, and those of MDA8 O$_3$ are 95.3, 96.4, 95.8, 101.3, 86.8, 92.0
and 89.6 µg m$^{-3}$, respectively. In terms of the annual average concentrations, PM$_{2.5}$ pollution has
improved significantly while O$_3$ pollution has not. In addition, PM$_{2.5}$ and O$_3$ pollution have clear
seasonal preferences, that is, PM$_{2.5}$ pollution tends to appear in cold months (November to February)
while O$_3$ pollution prefers to appear in warm months (April to August). High PM$_{2.5}$ concentrations
in cold months may be associated with the consumption of fossil fuels for heating and frequent
temperature inversion. Elevated O$_3$ concentrations in warm months are contributed to the high
temperature and strong sunlight.

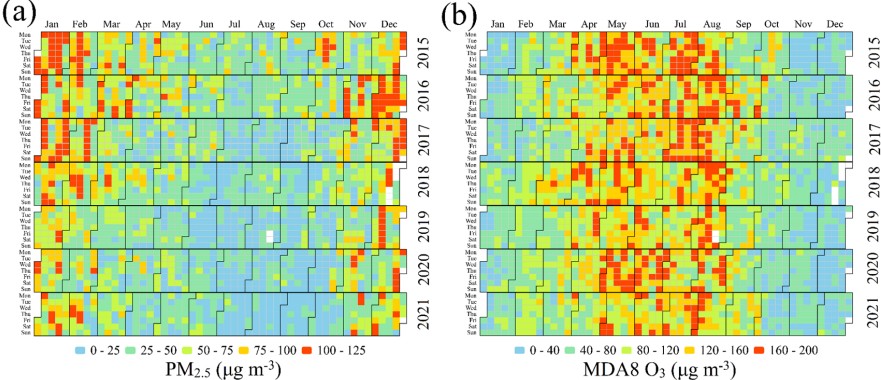


**Figure 2. Distribution of (a) daily average PM$_{2.5}$ and (b) MDA8 O$_3$ concentrations in Chengdu**
**from 2015 to 2021.**

**3.2 Premature mortality attributed to PM$_{2.5}$ and O$_3$**

Severe PM$_{2.5}$ and O$_3$ pollution are responsible for a considerable number of premature

mortalities in Chengdu. As shown in Table 4, the premature mortalities from ANAC attributed to
PM$_{2.5}$ exposure are 10596, 11647, 10154, 8942, 7993, 8298 and 8072 from 2015 to 2021, with an
annual average of 9386. The highest health risk among the diseases is from CVD with an annual





average of 2609, followed by COPD with an annual average of 1485 and RD with an annual average
of 1321. Due to urbanization and administrative division adjustment, Chengdu's population
increases by 24.2% from 2015 (16.853 million) to 2021 (20.938 million). In contrast, the total
premature mortalities attributed to $PM_{2.5}$ falls by 23.8% from 2015 (10596) to 2021 (8072). The
reduction in premature mortalities is mainly contributed to the reduction in $PM_{2.5}$ concentrations
since the annual average $PM_{2.5}$ concentrations decrease from 60.7 μg m$^{-3}$ in 2015 to 40.1 μg m$^{-3}$ in
2021 (Section 3.1), suggesting that air pollution control can bring huge health benefits.
Unlike the overall reduction in premature mortalities due to $PM_{2.5}$, the premature mortalities
due to $O_3$ fluctuate. The premature mortalities from ANAC attributed to $O_3$ exposure are 7657, 8025,
7870, 8556, 6367, 8300 and 7429 from 2015 to 2021, with an annual average of 7743, about 80%
of that attributed to $PM_{2.5}$ exposure. The total premature mortalities attributed to $O_3$ exposure in
2021 (7429) are only 3.0% lower than that in 2015 (7657). This is in line with the insignificant
reduction of $O_3$ concentrations in Chengdu from 2015 to 2021, indicating that $O_3$ pollution control
in Chengdu still has great potential and significance.

**Table 4.** Premature mortality from ANAC, CVD, RD and COPD attributed to $PM_{2.5}$ and $O_3$ exposure.

| Year | $PM_{2.5}$ | | | | MDA8 $O_3$ | | | |
|------|------|------|------|------|------|------|------|------|
|      | ANAC | CVD  | RD   | COPD | ANAC | CVD  | RD   | COPD |
| 2015 | 10596 | 2935 | 1485 | 1660 | 7657 | 1957 | 643 | 624 |
| 2016 | 11647 | 3231 | 1635 | 1832 | 8025 | 2053 | 672 | 653 |
| 2017 | 10154 | 2812 | 1422 | 1589 | 7870 | 2013 | 659 | 641 |
| 2018 | 8942 | 2490 | 1262 | 1424 | 8556 | 2191 | 715 | 696 |
| 2019 | 7993 | 2230 | 1131 | 1280 | 6367 | 1630 | 532 | 518 |
| 2020 | 8298 | 2313 | 1173 | 1325 | 8300 | 2121 | 696 | 677 |
| 2021 | 8072 | 2249 | 1140 | 1287 | 7429 | 1902 | 621 | 604 |


**3.3 Impacts of urbanization on $PM_{2.5}$ and $O_3$**
**3.3.1 Meteorological conditions in January and July**
In this study, January and July 2017 are chosen for simulations and analysis when $PM_{2.5}$ and



$O_3$ pollution are likely to occur (Figure 2). In January 2017, Chengdu experiences $PM_{2.5}$ pollution
for 23 out of 31 days with an average concentration of 128.5 μg m$^{-3}$. From the perspective of
atmospheric circulations, westerly winds prevail over Chengdu due to the large north-south
geopotential height gradient at 500 hPa (Figure 3a). However, the cold westerly winds from the
north are blocked by the Tibetan Plateau. Instead, a low-pressure system, called the Southwest
Vortex, appears to the left of Chengdu at 850 hPa (Figure 3b). Warm and humid southerly air flows
can reach Chengdu affected by this low-pressure system (Hu et al., 2021; Ning et al., 2018). The
dry air in the upper layer and moist air in the lower layer lead to a strong temperature inversion
appearing from 700 hPa to 500 hPa (Figure 4a and b). The blocking of clod air and the temperature
inversion make $PM_{2.5}$ pollution frequent during this period.

In July 2017, there are 19 days of $O_3$ pollution in Chengdu, and the average MDA8 $O_3$

concentration is 172.9 μg m$^{-3}$. At 500 hPa, Chengdu is dominated by strong high-pressure systems,
and thereby air temperature is high and wind speed is small (Figure 3c). The average $T_2$ is as high
as 28.6 °C while the average $WS_{10}$ is only 1.6 m s$^{-1}$ in July. These meteorological conditions are
conducive to the formation of $O_3$ pollution. Furthermore, the frequency and thickness of temperature
inversion in July are far less than those in January (Figure 4). Then $O_3$ can be well mixed within the
mixing layer during the daytime, which is an important way to maintain high surface $O_3$
concentrations (Aneja et al., 2000; Tang et al., 2017).



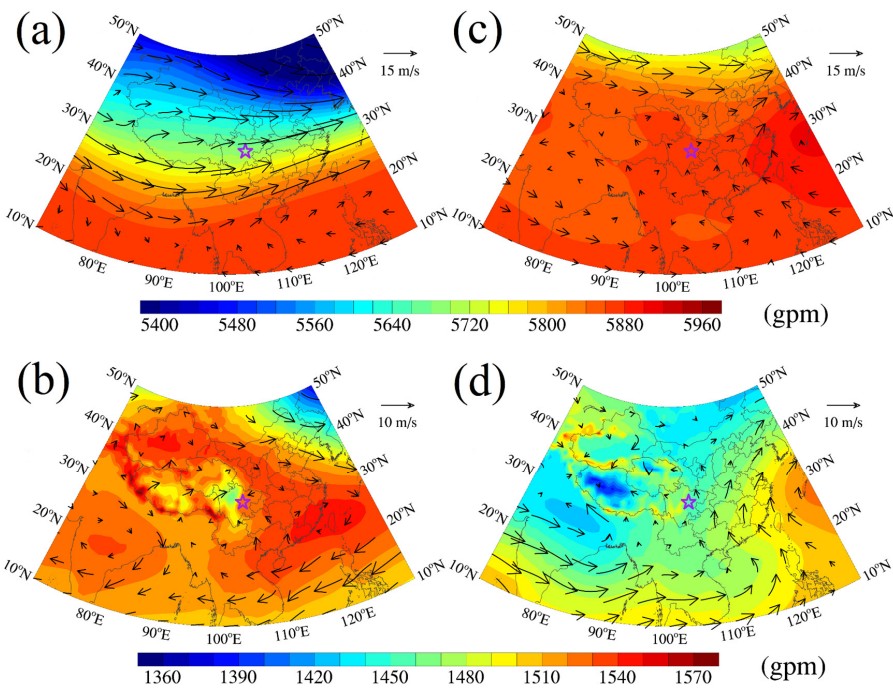

**Figure 3. The weather charts at (a) 500 hPa and (b) 850 hPa for January 2017. (c) and (d) are the same as (a) and (b), but for July 2017. The purple pentacles show the location of Chengdu. These weather charts are based on the NCEP global final analysis data.**





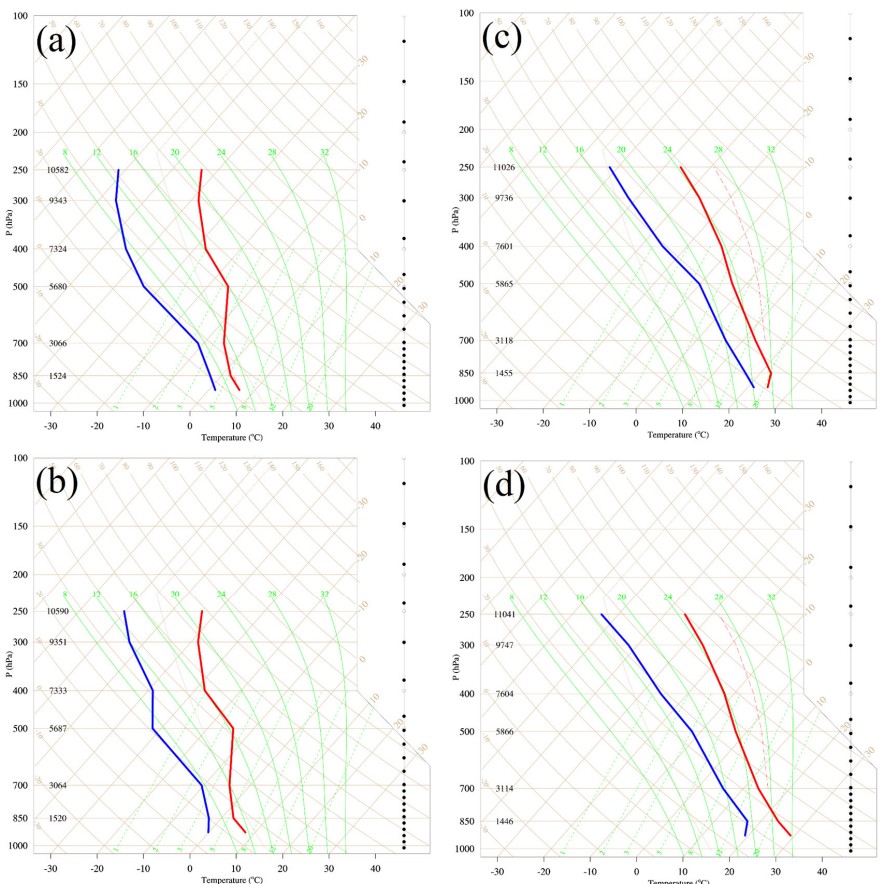

**Figure 4. The skew-T diagram at (a) 00:00 UTC and (b) 12:00 UTC in January 2017. (c) and (d) are the same as (a) and (b), but for July 2017. The red and blue lines indicate air temperature and dew point temperature, respectively.**

### 3.3.2 Evaluation of model performance

Simulated $PM_{2.5}$ concentrations, $O_3$ concentrations, air temperature, relative humidity and wind speed in baseline simulations are compared with the observations to verify the model performance (Figure 5). The magnitudes of simulated $PM_{2.5}$ and $O_3$ are reasonable with the small mean bias (MB) of 23.4 and 11.6 $\mu g\ m^{-3}$, respectively. The high correlation coefficients (COR) for $PM_{2.5}$ (0.44) and $O_3$ (0.77) indicate that simulations reproduce well the diurnal variation in pollutants. Therefore, the modeling results for $PM_{2.5}$ and $O_3$ are generally reasonable and acceptable. With regard to the meteorological factors, $T_2$ is well simulated with low MB (0.2 and 0.1 °C) and



high COR (0.76 and 0.70) values in both January and July. Our simulation underestimates RH to
some extent (the MB values are -14.3% and -4.8% in January and July, respectively), but the diurnal
variation of RH is well represented (the COR values are 0.54 and 0.64). As for $WS_{10}$, poor
simulation results are predictable in the case of low wind and complex terrain. The $WS_{10}$ in the
model is overestimated (the MB values are 1.3 and 1.7 m s$^{-1}$), which may be related to the unresolved
terrain features by the default surface drag parameterization causing an overestimate of wind speed
in particular at low values (Jimenez and Dudhia, 2012). Due to the small change in weak wind, the
COR for $WS_{10}$ is not high. In general, the WRF-Chem model using our configuration has a good
capability in simulating air pollutants and meteorological factors in Chengdu, and thereby the
simulations can be used for subsequent analysis.



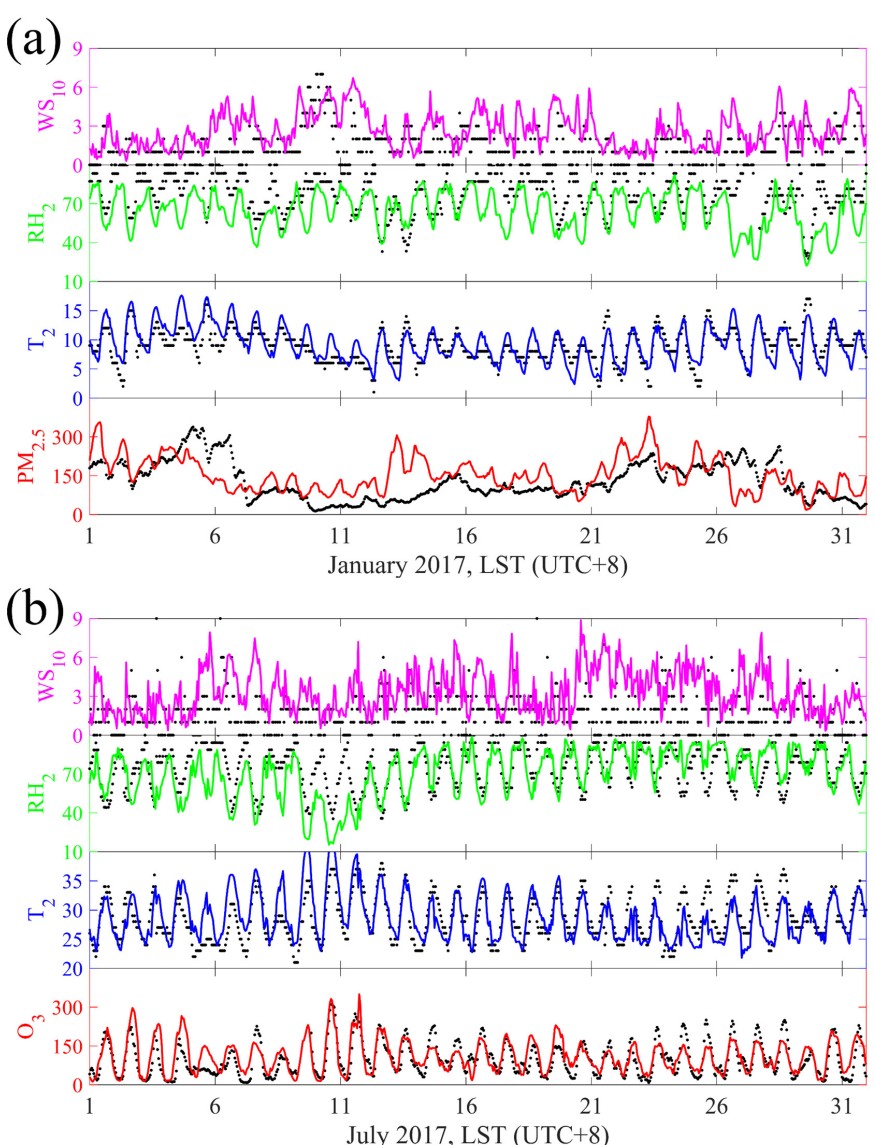

**Figure 5. Times series of PM$_{2.5}$, O$_3$, T$_2$, RH and WS$_{10}$ for observations (black dots) and baseline simulations (colored lines).**

### 3.3.3 Spatiotemporal variations in PM$_{2.5}$ and O$_3$

Figure 6 shows the January-averaged spatiotemporal distribution of PM$_{2.5}$ in Jan_Base simulation. PM$_{2.5}$ has a diurnal variation with high concentration at night and low concentration at





noon, which is contrary to the diurnal variation of the boundary layer height. At night, the boundary
layer height is usually low. As a consequence, $PM_{2.5}$ is trapped and maintained on the ground. At
noon, turbulence in the convective boundary layer can dilute $PM_{2.5}$ concentrations through vertical
mixing, resulting in low $PM_{2.5}$ concentrations at surface. Chengdu is on the east side of the Tibetan
Plateau, with a large elevation drop exceeding 3000 m over a short horizontal distance (Figure 1a).
In this case, the mountain-valley breezes can easily develop in winter when atmospheric conditions
are usually stagnant, and are crucial for $PM_{2.5}$ in Chengdu. The $PM_{2.5}$ pollution zone tends to appear
in the converging airflows associated with the mountain breezes and can spread hundreds of
kilometers (Figure 6a-d). Driven by the near-surface northeasterly winds, $PM_{2.5}$ is uplifted over the
windward slope of the Tibetan Plateau (Figure 6e-h). Then the uphill airflows are restrained and
overturned below 3 km, forming a vertical secondary circulation over Chengdu. Governed by the
secondary circulation forced by the complex terrain, the southwesterly winds at 3 km can transport
$PM_{2.5}$ downward, which could replenish the surface $PM_{2.5}$ and facilitate the accumulation and
maintenance of surface $PM_{2.5}$.

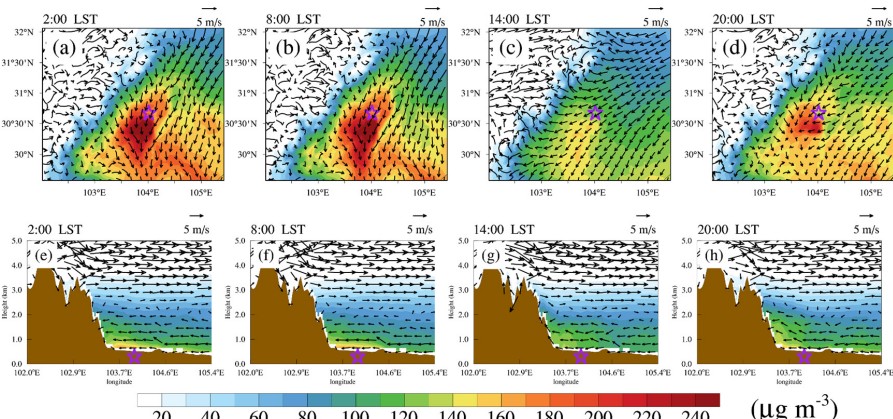


**Figure 6. (a-d) Spatial distributions and (e-h) east-west vertical cross sections of $PM_{2.5}$ with**
**wind fields at 2:00, 8:00, 14:00 and 20:00 LST (LST is UTC+8h) in Jan_Base simulation.**
**Purple pentacles show the location of Chengdu. Brown-shaded areas represent the terrain.**

In terms of $O_3$, it exhibits strong diurnal variation with an afternoon maximum and an early

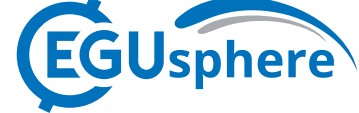



morning minimum (Figure 7a-d). After sunrise, the nocturnal residual layer is destroyed while the
convective boundary layer begins to form as the surface is heated. This leads to downward mixing
of $O_3$ from aloft (Hu et al., 2018; Zhan and Xie, 2022). Meanwhile, $O_3$ is also generated by
photochemical reactions between volatile organic compounds (VOCs) and $NO_x$ in the presence of
sunlight. As a consequence, $O_3$ concentrations increase rapidly from morning to noon (Figure 7b,
7c, 7f and 7g). By noon, $O_3$ can be well mixed within the convective boundary layer via strong
turbulence. Strong photochemical production and well mixing cause high $O_3$ concentrations until
late afternoon (Figure 7c, 7d, 7g and 7h). Thereafter, $O_3$ production decreases since the intensity of
sunlight diminishes. After sunset, $O_3$ concentrations decrease substantially due to surface deposition
and nitrogen oxide titration ($O_3 + NO \rightarrow O_2 + NO_2$) and reach their minimum in the early morning
(Figure 7a, 7d, 7e and 7h). But $O_3$ in the nocturnal residual layer is still at a high level with values
more than 140 $\mu g\ m^{-3}$. Compared with the Jan_Base simulation, the secondary circulation forced by
the complex terrain is not obvious. In addition, $O_3$ with a concentration of about 100 $\mu g\ m^{-3}$ has
always existed over the Tibetan Plateau, but $PM_{2.5}$ concentrations there are quite low. This indicates
that the background concentration of $O_3$ is much higher than that of $PM_{2.5}$, which may pose a huge
challenge to $O_3$ pollution control.

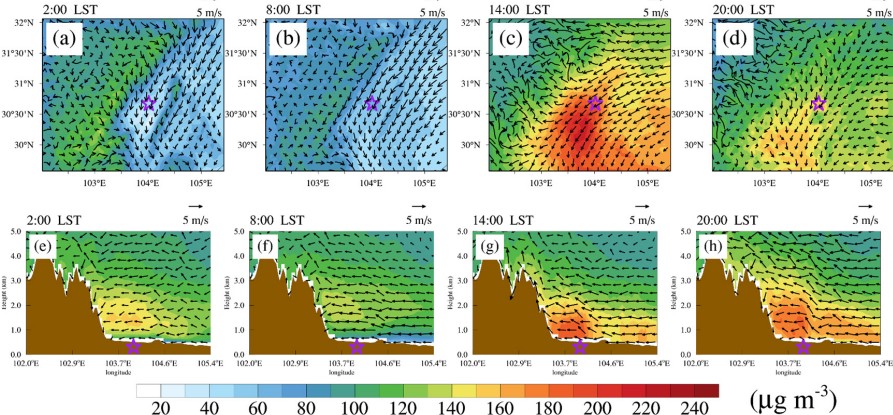


**Figure 7. (a-d) Spatial distributions and (e-h) east-west vertical cross sections of $O_3$ with wind**
**fields at 2:00, 8:00, 14:00 and 20:00 LST (LST is UTC+8h) in July_Base simulation. Purple**
**pentacles show the location of Chengdu. Brown-shaded areas represent the terrain.**






### 3.3.4 Impacts of urban expansion on $PM_{2.5}$ and $O_3$


Modification of urban land use changes surface dynamic and thermal characteristics, and
thereby affects the transportation and dispersion of air pollutants. Figure 8 shows the differences in
$PM_{2.5}$ between Jan_Base and Jan_noCD simulations (Jan_Base minus Jan_noCD). Results show
that surface $PM_{2.5}$ concentrations in Jan_Base simulation are lower at all times compared with
Jan_noCD simulation, with the monthly average decreasing by 11.7 $\mu g\ m^{-3}$ (Figure 8a-d). Moreover,
the decrease in $PM_{2.5}$ is larger during the nighttime than during the daytime. Specially, surface $PM_{2.5}$
concentrations decrease by 15.0 $\mu g\ m^{-3}$ at 2:00 LST and 3.2 $\mu g\ m^{-3}$ at 14:00 LST. The decrease in
surface $PM_{2.5}$ is mainly attributed to the modification of the boundary layer height. Urban land use
can enhance surface heating leading to an increase in air temperature, known as the urban heat island.
The vertical air movement is then enhanced by the warming up of surface air temperature, resulting
in an increase in the boundary layer height, which facilitates the vertical diffusion of surface $PM_{2.5}$.
$PM_{2.5}$ concentrations increase by 2-6 $\mu g\ m^{-3}$ in the upper boundary layer (~1 km above the surface)
(Figure 8e-h), further confirming this point.

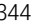
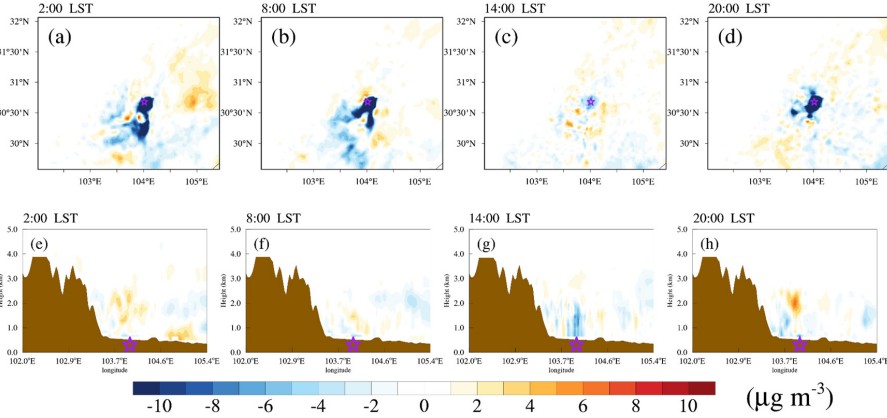


**Figure 8. (a-d) Spatial distributions and (e-h) east-west vertical cross sections of the differences**
**in $PM_{2.5}$ between Jan_Base and Jan_noCD simulations (Jan_Base minus Jan_noCD). Purple**
**pentacles show the location of Chengdu. Brown-shaded areas represent the terrain.**




O₃ is a secondary air pollutant that is not only related to meteorological conditions but also to
its precursors (VOCs and $NO_x$). Due to the increase in upward air movement and boundary layer
height induced by urban land use, $PM_{2.5}$ and $NO_x$ concentrations decrease near the surface but
increase in the upper boundary layer (Liao et al., 2015; Zhu et al., 2017). The decrease in $NO_x$ near
the surface results in an increase in surface O₃ at night since the $NO_x$ titration is weakened (Figure
9a and d). Although the elevated boundary layer dilutes O₃ concentrations to some extent, the
nighttime O₃ concentrations are mainly dominated by chemical effects and eventually increase by a
maximum of 25.8 $\mu g\ m^{-3}$. During the daytime, the increased air temperature is conducive to the
photochemical production of O₃, and the well-developed convective boundary layer favors vertical
mixing of O₃. O₃ concentrations will also increase (Figure 9b and c), with the value of 4.4 $\mu g\ m^{-3}$ at
14:00 LST in Chengdu. Finally, MDA8 O₃ concentrations in July can increase by 10.6 $\mu g\ m^{-3}$ due
to the effects of urban expansion.

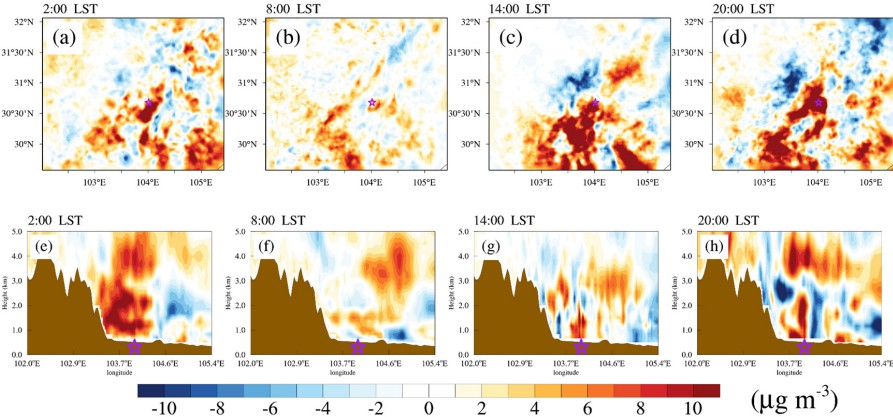


**Figure 9. (a-d) Spatial distributions and (e-h) east-west vertical cross sections of the differences**
**in O₃ between July_Base and July_noCD simulations (July_Base minus July_noCD). Purple**
**pentacles show the location of Chengdu. Brown-shaded areas represent the terrain.**

**3.3.5 Impacts of anthropogenic emissions on $PM_{2.5}$ and O₃**
Air pollutants become part of the air once released, and they do not have a direct effect on
airflows like temperature or radiation. Therefore, the impacts of anthropogenic emissions are more

 

intuitive than urban expansion. Figure 10 shows the differences in PM$_{2.5}$ between Jan_Base and
Jan_noEmi simulations (Jan_Base minus Jan_noEmi). PM$_{2.5}$ concentrations in Jan_Base simulation
are significantly higher than those in Jan_noEmi simulations, with the monthly average
concentration increasing by 26.6 μg m$^{-3}$, more than twice the difference between Jan_Base and
Jan_noCD simulations. Furthermore, the increase in PM$_{2.5}$ appears throughout the boundary layer
(Figure 10e-h) and can extend downstream for hundreds of kilometers (Figure 10a-d), indicating
that reducing anthropogenic emissions is an effective way to reduce PM$_{2.5}$ concentrations.

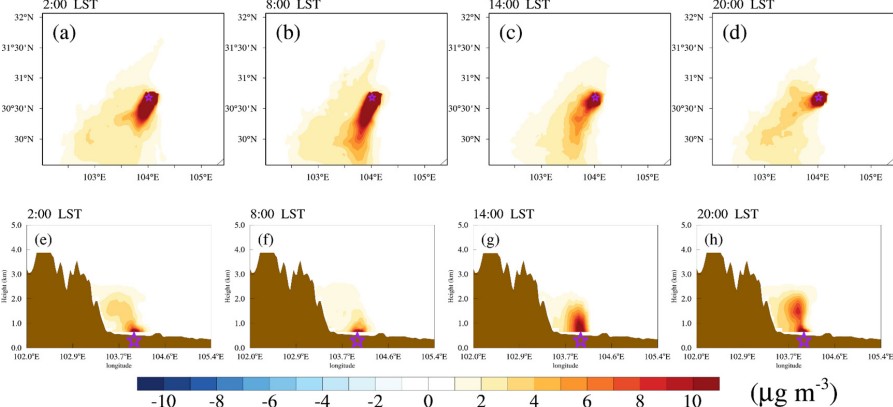


**Figure 10. (a-d) Spatial distributions and (e-h) east-west vertical cross sections of the**
**differences in PM$_{2.5}$ between Jan_Base and Jan_noEmi simulations (Jan_Base minus**
**Jan_noEmi). Purple pentacles show the location of Chengdu. Brown-shaded areas represent**
**the terrain.**

As for O$_3$, O$_3$ concentrations in July_Base simulation are 5.8 μg m$^{-3}$ higher than those in

July_noEmis simulation at 14:00 LST due to the abundance of O$_3$ precursors (Figure 11c). However,
O$_3$ concentrations decrease by 3.4 μg m$^{-3}$ at 2:00 LST (Figure 11a). This phenomenon may be related
to the non-linear sensitivity of O$_3$ to VOCs and NO$_x$ precursor emissions. O$_3$ formation regimes are
often classified into VOC-limited, NO$_x$-limited and transition regimes depending on the ratio of
VOCs and NO$_x$ (Jin et al., 2020; Lu et al., 2019). At low VOC/NO$_x$ ratios (VOC-limited regime,
usually in urban areas), reducing the concentrations of NO$_x$ would lead to an increase in O$_3$



formation. Apart from the amount of anthropogenic emissions, a reasonable emission reduction path
is also necessary to alleviate O$_3$ pollution. Since O$_3$ concentrations increase during the daytime,
MDA8 O$_3$ concentrations in July_Base simulation are still 4.8 μg m$^{-3}$ higher than those in
July_noEmis simulation.

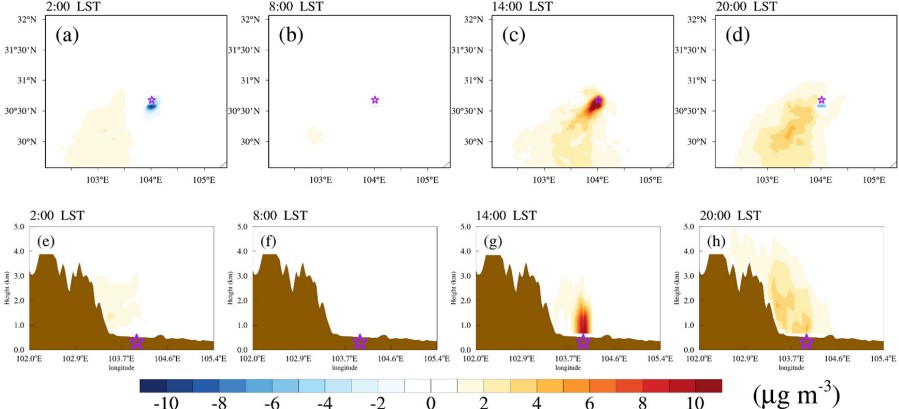


**Figure 11. (a-d) Spatial distributions and (e-h) east-west vertical cross sections of the**
**differences in O$_3$ between July_Base and July_noEmi simulations (July_Base minus**
**July_noEmi). Purple pentacles show the location of Chengdu. Brown-shaded areas represent**
**the terrain.**

**3.4 Health risks caused by urbanization**
According to the above results, urban expansion can decrease surface PM$_{2.5}$ by 11.7 μg m$^{-3}$ but
increase MDA8 O$_3$ by 10.6 μg m$^{-3}$. On the other hand, anthropogenic emissions can increase surface
PM$_{2.5}$ by 26.6 μg m$^{-3}$ and MDA8 O$_3$ by 4.8 μg m$^{-3}$. We then calculate the premature mortalities
under different simulation scenarios to access the health risks from changes in PM$_{2.5}$ and O$_3$
concentrations. As shown in Figure 12, the premature mortalities from ANAC, CVD, RD and COPD
due to PM$_{2.5}$ decrease by 182 (6.9%), 47 (6.5%), 23 (6.4%) and 24 (6.1%) in January 2017 with the
existence of Chengdu. While anthropogenic emissions in Chengdu increase premature mortalities
from ANAC, CVD, RD and COPD due to PM$_{2.5}$ by 424 (16.0%), 111 (15.4%), 55 (15.2%) and 56
(14.3%), respectively. With regard to O$_3$, premature mortalities from the O$_3$-induced diseases all



increase when urban land use and anthropogenic emissions are taken into account. Urban expansion
leads to an increase of premature mortalities from ANAC, CVD, RD and COPD due to $O_3$ by 203
(9.5%), 51 (9.4%), 18 (10.0%) and 17 (9.7%) in July 2017, respectively. When anthropogenic
emissions in Chengdu are turned on, premature mortalities from ANAC, CVD, RD and COPD due
to $O_3$ can increase by 87 (4.1%), 22 (4.1%), 8 (4.4%) and 7 (4.0%), respectively. In summary,
affected by urban expansion and anthropogenic emissions, changes in total premature mortalities
due to $PM_{2.5}$ are –6.9% and 16.0%, due to $O_3$ are 9.5% and 4.1%. The effects of urban expansion
on health risks are in the same order (1/2 to 2 times) as those induced by anthropogenic emissions.

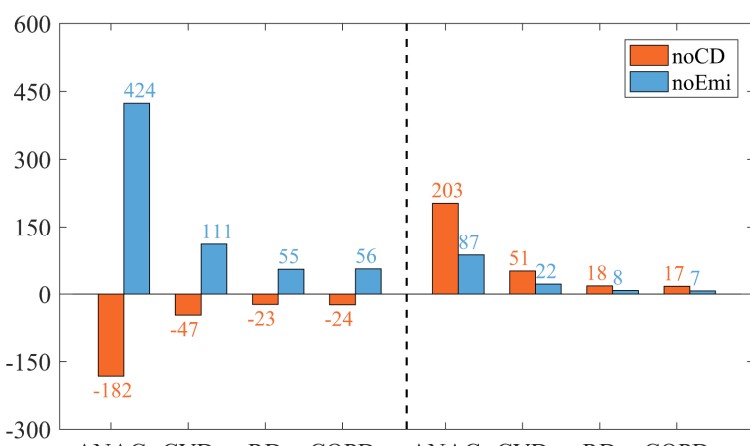


**Figure 12. Differences in premature mortality from ANAC, CVD, RD and COPD attributed**
**to $PM_{2.5}$ (left of dotted line) and $O_3$ (right of dotted line) exposure between baseline and**
**sensitivity simulations. The red bars are the differences between Jan_Base and Jan_noCD**
**simulations (Jan_Base minus Jan_noCD), and the differences between July_Base and**
**July_noCD simulations (July_Base minus July_noCD). The blue bars are the differences**
**between Jan_Base and Jan_noEmi simulations (Jan_Base minus Jan_noEmi), and the**
**differences between July_Base and July_noEmi simulations (July_Base minus July_noEmi).**

**4 Conclusions**
With the development in urbanization, urban land use and anthropogenic emissions increase,
which affects urban air quality and then health risks of air pollutants. In this study, the impacts of





434 urbanization on air quality and the corresponding health risks in Chengdu, a highly urbanized city

435 with severe air pollution and complex terrain, are quantified. Management of urban air pollution is

436 usually achieved by reducing anthropogenic emissions. So the impacts of urban expansion are

437 further compared with anthropogenic emissions on health risks.

438  Chengdu is suffering from severe $PM_{2.5}$ and $O_3$ pollution in recent years. There are 97, 101, 68,

439 53, 33, 43 and 37 $PM_{2.5}$ pollution episodes, and 61, 48, 42, 40, 26, 50 and 27 $O_3$ pollution episodes

440 in Chengdu from 2015 to 2021. Severe $PM_{2.5}$ and $O_3$ pollution pose huge health risks. The annual

441 premature mortalities from ANAC, CVD, RD and COPD due to $PM_{2.5}$ are 9386, 2609, 1321 and

442 1485, due to $O_3$ are 7743, 1981, 648 and 630. $PM_{2.5}$ and $O_3$ pollution have different seasonal

443 preferences. Due to the secondary circulation driven by complex terrain and the frequent

444 temperature inversion, $PM_{2.5}$ pollution tends to appear in cold months (November to February).

445 While $O_3$ pollution is likely to occur in warm months (April to August) because of high temperature

446 and strong sunlight dominated by high-pressure systems. $PM_{2.5}$ has a diurnal variation with high

447 concentrations at night but low concentrations at noon affected by the boundary layer height. $O_3$

448 exhibits strong diurnal variation with an afternoon maximum and an early morning minimum, which

449 is related to photochemical reactions during the daytime and nitrogen oxide titration at night.

450  The urban land use of Chengdu is replaced by cropland in the WRF-Chem model to examine

451 the impacts of urban expansion. Urban expansion leads to an increase in air temperature and

452 boundary layer height, and decreases surface $PM_{2.5}$ concentrations by 11.7 $\mu g\ m^{-3}$ in January 2017.

453 As for $O_3$, the surface concentration increases by 4.4 $\mu g\ m^{-3}$ at noon due to stronger photochemical

454 reactions and better vertical mixing, and increases by 25.8 $\mu g\ m^{-3}$ at midnight since the $NO_x$ titration

455 is weakened. MDA8 $O_3$ finally increases by 10.6 $\mu g\ m^{-3}$ in July 2017 when urban land use is taken

456 into account. In this case, the premature mortalities from ANAC, CVD, RD and COPD attributed

457 to $PM_{2.5}$ exposure decrease by 182 (6.9%), 47 (6.5%), 23 (6.4%) and 24 (6.1%), attributed to $O_3$

458 exposure increase by 203 (9.1%), 51 (9.4%), 18 (10.0%) and 17 (9.7%). Anthropogenic emissions

459 increase surface $PM_{2.5}$ significantly with monthly average concentration increasing by 26.6 $\mu g\ m^{-3}$,

460 more than twice the difference caused by urban expansion. Due to the non-linear sensitivity of $O_3$

461 to its precursors, $O_3$ concentrations increase during the daytime but decrease at night. In particular,

462 $O_3$ concentrations increase by 5.8 $\mu g\ m^{-3}$ at 14:00 LST but decrease by 3.4 $\mu g\ m^{-3}$ at 2:00 LST with

463 anthropogenic emissions in Chengdu. Since $O_3$ concentrations in daytime are much higher than



those at night, MDA8 $O_3$ concentrations still increase by 4.8 μg m$^{-3}$. As a consequence, the
premature mortalities from ANAC, CVD, RD and COPD attributed to $PM_{2.5}$ exposure increase by
424 (16.0%), 111 (15.4%), 55 (15.2%) and 56 (14.3%), attributed to $O_3$ exposure increase by 87
(4.1%), 22 (4.1%), 8 (4.4%) and 7 (4.0%).
Our results show that the impacts of urban expansion are in the same order (1/2 to 2 times) as
those induced by emissions growth on air pollutants. Although the focus of air quality management
is traditionally to regulate emissions, urban planning is an ancillary option and should also be
considered in future air pollution strategies.

***Data Availability Statement.***
Air quality monitoring data are acquired from the official NEMC real-time publishing platform
(http://106.37.208.233:20035/). Meteorological data are taken from the website of the University of
Wyoming (http://weather.uwyo.edu/). The NCEP global final analysis data were taken from the
NCEP (https://doi.org/10.5065/D6M043C6/). The MEIC data are accessible at
http://meicmodel.org/. These data can be downloaded for free as long as you agree to the official
instructions.

***Author contributions.***
CZ and MX had the original ideas, designed the research, collected the data and prepared the original
draft. CZ did the numerical simulations and carried out the data analysis. MX acquired financial
support for the project leading to this publication. HL, BL and ZW collected the data. TW, BZ, ML
and SL reviewed the initial draft and checked the language of the original draft.

***Competing interests.***
The contact author has declared that neither they nor their co-authors have any competing interests.

***Acknowledgements.***
We are grateful to NEMC for the air quality monitoring data, to NCDC for the meteorological data,
to NCEP for global final analysis fields and to Tsinghua University for the MEIC inventories. The
numerical calculations were performed on the Blade cluster system in the High Performance

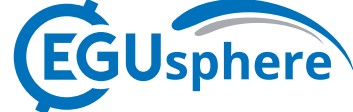

Computing and Massive Data Center (HPC&MDC) of School of Atmospheric Sciences, Nanjing
University.

***Financial support.***
This work was supported by the open research fund of Chongqing Meteorological Bureau (KFJJ-
201607), the Natural Science Foundation of Chongqing (cstc2021jcyj-msxmX1007),
the Chongqing Science and Technology Commission technology innovation and application
demonstration project (cstc2018jszx-zdyfxmX0003) and Innovation Team Fund of Southwest
Regional Meteorological Center, China Meteorological Administration.

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
