# Peer review of "Impacts of urbanization on air quality and related health risks in"

_EGUsphere, 2022_

## Author Response (AR1)

**Response to Reviewers**

Thank you for the reviewers' comments on our manuscript entitled "Impacts of urbanization on air quality and related health risks in a city with complex terrain" (egusphere-2022-486). These constructive comments are all valuable for revising and improving our manuscript. We study these comments carefully and have made correction as requested. Here are point-by-point responses (in blue color), and the changes are reflected in the revised manuscript (in red color). The line numbers in the authors' responses are obtained from the revised manuscript, in which all the revisions have been accepted.

**Anonymous Referee #1:**

The paper presents WRF-Chem simulations to analyze changes in $PM_{2.5}$ and $O_3$ concentrations due to urbanization effects in the Chinese city of Chengdu. Two months are simulated, specifically January for $PM_{2.5}$ and July for $O_3$, and sensitivity experiments are performed for each of the two simulations to evaluate the impact of land-use changes associated with urban expansion and the impact of anthropogenic emissions. The authors also estimate the changes in premature mortality associated with the changes in air pollution concentrations. The study is well designed to differentiate between the meteorological changes associated with urbanization caused by land-use changes and the changes in emissions and is, in my opinion, a nice example of how a modeling study can be used to disentangle different factors. The paper is also overall very well written.

I have, however, some concerns related to the lack of information about the boundary layer depth and vertical structure despite its supposedly large impact on the results (general comment 1) and the seemingly strong overestimation of $PM_{2.5}$ in the model (general comment 2).

*Response:* We appreciate the time and efforts you have dedicated to providing valuable comments on our manuscript. The manuscript has been modified and much improved based on those constructive comments. As for the two general comments mentioned by the reviewer, in the revised manuscript, we have added more information on the planetary boundary layer (general comment 1), and we have run new simulations on $PM_{2.5}$ (general comment 2). A point-by-point response can be found below.

**General comments**

1. The text refers repeatedly to the boundary layer height to explain, e.g., the temporal variations in PM$_{2.5}$ (e.g., line 287, line 308), which makes of course perfect sense. These statements are, however, rather phrased as assumptions or hypotheses and the structure and development of the boundary layer is never discussed or even evaluated to make sure that the model actually captures this crucial factor correctly. In contrast to the authors' statement (line 266), I would say that the model does not reproduce the diurnal cycle of PM$_{2.5}$ well (Fig. 5). The correlation coefficient is not that high and the model also generally overestimates the observed values during most of the time. This could potentially be connected to a poor representation of the boundary layer.

*Response:* We thank the reviewer for pointing out the inadequacies of our simulations. We have tried our best to revise our manuscript based on the following comments.

a. I would recommend to start by comparing vertical profiles in the model with the radiosounding data to determine how well the model actually captures the vertical structure of the boundary layer. This is also commonly done before comparing the model output to surface observations to evaluate the larger scale first. This may even be helpful in better understanding the model performance for surface wind to see whether the large-scale flow is actually represented correctly.

*Response:* Thanks for the constructive comment. We take your suggestion and compare vertical profiles in the model with the sounding observations in Section 3.3.2 Evaluation of model performance. Please see lines 300-304 in the revised manuscript. As shown in Figure R1, the WRF-Chem model can roughly capture the vertical structure of the tropospheric atmosphere and the large-scale flow, indicating that the simulated vertical structure of the boundary layer is reasonable.

[Figure]

**Figure R1. The skew-T diagram at (a) 00:00 UTC and (b) 12:00 UTC in January 2017. (c) and (d) are the same as (a) and (b), but for July 2017. The red and blue lines are the simulated air temperature and dew point temperature, while the red and blue points are the sounding temperature and dew point temperature. These results are monthly averages.**

b. Can it be shown that the temporal development of the boundary-layer depth and vertical structure can explain the modeled development in PM$_{2.5}$ and O$_3$ as discussed in the paper? Similarly, the text also hypothesizes about the impact of the urban heat island (line 339: "Urban land use can enhance surface heating leading to an increase in air temperature … resulting in and increase in the boundary layer height"). Can this actually be shown in the simulations?

*Response:* Thanks for the constructive comment. In the revised Section 3.3.3, the temporal-vertical cross sections of boundary layer height, PM$_{2.5}$ (Figure 7a) and O$_3$ (Figure 8a) are given to explain their diurnal evolution. O$_3$ concentrations and boundary layer height are high during the day and low at night, while PM$_{2.5}$ concentrations are high at night and low during the day. Based on new Figure 7 and 8, we deeply reorganize the language in Section 3.3.3 to revisit the spatiotemporal

variations in $PM_{2.5}$ and $O_3$. Please see lines 328-343 for $PM_{2.5}$ and lines 352-371 for $O_3$.

Urban land use can lead to an increase in air temperature due to increased sensible heat, and an increase in boundary layer height due to higher air temperature, which will decrease surface $PM_{2.5}$ concentrations but increase surface $O_3$ concentrations. This can be confirmed by our simulation results (Figure R2 and R3), and has also been reported by other scholars (e.g. Jiang et al., 2008; Li et al., 2019). However, the increases in air temperature and boundary layer height were not shown in original Figure 8 and 9 as we only gave the differences in $PM_{2.5}$ and $O_3$ concentrations in these two figures. In the revised manuscript, we add supporting information containing the relevant content. Please see lines 18-27 in the supporting information.

[Figure]

**Figure R2. Horizontal distributions of differences in (a-d) 2 m air temperature, (e-h) boundary layer height and (i-j) $PM_{2.5}$ concentrations between Jan_Base and Jan_noCD (Jan_Base − Jan_noCD). The purple dots represent the locations of Chengdu.**

[Figure]

**Figure R3. Horizontal distributions of differences in (a-d) 2 m air temperature, (e-h) boundary layer height and (i-j) O₃ concentrations between Jan_Base and Jan_noCD (Jan_Base − Jan_noCD). The purple dots represent the locations of Chengdu.**

References

Jiang, X., Wiedinmyer, C., Chen, F., Yang, Z.-L., and Lo, J. C.-F.: Predicted impacts of climate and land use change on surface ozone in the Houston, Texas, area, Journal of Geophysical Research, 113, 10.1029/2008jd009820, 2008.

Li, Y., Zhang, J., Sailor, D. J., and Ban-Weiss, G. A.: Effects of urbanization on regional meteorology and air quality in Southern California, Atmospheric Chemistry and Physics, 19, 4439-4457, 10.5194/acp-19-4439-2019, 2019.

c. Have the authors performed any sensitivity tests with different PBL parameterizations? They can have a large impact on the boundary-layer structure and depth, which could potentially have a large impact on the results.

*Response:* Thanks for the constructive comment. Yes, we investigated and tested different PBL parameterizations before we performed numerical simulations. Table R1 shows the statistical metrics between simulations and observations for MYJ and YSU schemes, the two most common PBL parameterizations in WRF. Overall, the MYJ scheme performed better than the YSU scheme, especially in the overestimation of $PM_{2.5}$ concentration. In some of the literature we surveyed (Hu and Wang, 2021; Shu et al., 2021), the MYJ scheme was recommended for complex terrain conditions. Thanks to reviewer #2 for **Comment 9** and the references, we add a comparison of our model performance with previous studies to demonstrate the robustness of model results. Please see lines 309-311 and 314-318 for details. By the way, the references provided by reviewer #2 also use the MYJ scheme.

**Table R1.** Statistical metrics between observations and simulations.

| Variables | $\overline{O}$ | MYJ scheme | | | | YSU scheme | | | |
|---|---|---|---|---|---|---|---|---|---|
| | | $\overline{S}$ | MB | RMSE | COR | $\overline{S}$ | MB | RMSE | COR |
| $PM_{2.5}$ ($\mu g\ m^{-3}$) | 128.5 | 152.5 | 23.4 | 77.9 | 0.44 | 209.9 | 81.4 | 155.2 | 0.34 |
| $T_2$ (℃) | 9.0 | 9.2 | 0.2 | 2.0 | 0.76 | 8.7 | –0.3 | 2.2 | 0.76 |
| $TD_2$ (℃) | 5.2 | 2.4 | –2.8 | 4.1 | 0.44 | 2.0 | –3.2 | 4.4 | 0.50 |
| $WS_{10}$ ($m\ s^{-1}$) | 1.3 | 2.7 | 1.3 | 2.0 | 0.36 | 2..4 | 1.1 | 1.8 | 0.32 |

Note. $\overline{O}$, $\overline{S}$, MB, RMSE and COR are the average of observations, the average of simulations, the mean bias, the root mean square error and the correlation coefficient, respectively. $T_2$, $TD_2$ and $WS_{10}$ are 2-m air temperature, 2-m dew point temperature and 10-m wind speed, respectively.

References

Hu, Y. and Wang, S.: Formation mechanism of a severe air pollution event: A case study in the Sichuan Basin, Southwest China, Atmospheric Environment, 246, 10.1016/j.atmosenv.2020.118135, 2021.

Shu, Z., Liu, Y., Zhao, T., Xia, J., Wang, C., Cao, L., Wang, H., Zhang, L., Zheng, Y., Shen, L., Luo, L., and Li, Y.: Elevated 3D structures of $PM_{2.5}$ and impact of complex terrain-forcing circulations on heavy haze pollution over Sichuan Basin, China, Atmos. Chem. Phys., 21,

9253-9268, 10.5194/acp-21-9253-2021, 2021.

2. As mentioned above, the model seems to overestimate PM$_{2.5}$ relatively strongly, with a mean bias of 23.4 μg m$^{-3}$, which the authors consider to be small (line 264). This is almost the same magnitude as the monthly mean change related to anthropogenic emissions (26.6 μg m$^{-3}$), which the authors call significant (line 373), and twice the monthly mean change related to urban expansion. During individual days, the overestimation reaches even values of up to 200 μg m$^{-3}$ (estimated from Fig. 5). I am somewhat concerned about the value of the results from the sensitivity tests considering the equally large model error and maybe that some processes affecting PM$_{2.5}$ concentrations are not simulated correctly.

*Response:* Thanks for the constructive comment. The quality of atmospheric chemical models depends on many factors, of which accurate meteorological, chemical and emission processes are critical. These three requirements are seldom uniformly met, so it is often unclear whether inadequacies in air quality models are due to meteorological, chemical or emissions (or all three) weaknesses. A simple common judgment is that when the normalized mean bias (NMB) is within ±15%, the simulation of pollutants is acceptable. Thanks for pointing out that our model overestimated PM$_{2.5}$ concentrations and thanks to reviewer #2 for the reminder to check emissions data. Considering that emissions are in most cases the major factor limiting the accuracy of air quality forecasts. We check the anthropogenic emissions in the Multi-resolution Emission Inventory for China (MEIC) and empirically cut the anthropogenic emissions by about 20%. The new NMB value of PM$_{2.5}$ is 9.9%, and the mean bias value of PM$_{2.5}$ drops from 23.4 μg m$^{-3}$ to 12.7 μg m$^{-3}$. In the revised manuscript, all results on PM$_{2.5}$ are from new simulations.

As for the overestimation or underestimation of pollution in some periods, we believe that this may be related to meteorological weakness, especially for wind flows. In practice, it is often difficult to ensure that the meteorological conditions are correct for each time period in a long-term simulation. Since the effects of urbanization are long-term, and the health risks of pollutants are statistically significant, we focus on the entire simulation period to avoid some special situations, such as cold fronts and setting off fireworks. Furthermore, the differences in air pollutant concentrations between baseline and sensitivity simulations are used when assessing the impacts of

urbanization on health risks of air pollutants. Overestimation or underestimation in some periods may not significantly change the differences in health risks over the entire time period. For example, both the original and new simulations show that urban land use could decrease premature mortalities due to $PM_{2.5}$ by about 6.9%.

3. The paper reports very precise numbers for the changes caused by urbanization, e.g., the number of premature deaths resulting from $PM_{2.5}$ (9386, 2609, 1321, and 1485 from ANAC, CVD, RD, and COPD) and the corresponding decreases of 424, 111, 55, and 56 without anthropogenic emissions. Considering that the equations given in Section 2.3 can likely provide only estimates and that I assume there must be large spatial variations in population density and air pollution throughout the city as well, which will affect the total premature deaths, I find the precision of these numbers somewhat misleading. Similarly, in the introduction a paper is cited by Liao et al. (2015) listing temperature and $PM_{10}$ concentrations caused by urbanization with a resolution of 0.1ºC and 0.1 µg $m^{-3}$ while providing a range of more than 1 ºC and more than 40 µg $m^{-3}$ and no information about the magnitude of the urban expansion.

*Response:* Thanks for the constructive comment. We agree that it is inappropriate to report precise numbers of premature deaths due to $PM_{2.5}$ and $O_3$. The formulas for calculating the health risks of air pollutants are derived from epidemiology and the results are statistically significant. Premature deaths ($\Delta M$) depend on the exposed population ($Pop$), the baseline mortality rate ($y_0$), the concentration-response function ($\beta$), the pollutant concentration ($C$) and the threshold concentration ($C_0$). Among these factors, the greatest uncertainty comes from $\beta$, which reflects the relationships between exposure and attributable deaths. Therefore, $\beta$ at 95% confidence intervals (95% CI) is used in most studies. In the revised manuscript, premature deaths are presented based on $\beta$ at 95% CI. The spatial distribution of population and pollutant concentrations undoubtedly affects the number of premature deaths. Some scholars use gridded population data to obtain a fine distribution of premature deaths (Li et al., 2020; Liu et al., 2018). But this is mostly the case for the population distribution in a particular year. The output of the numerical model is itself grided, but pollution monitoring stations rarely fall on a grid. Therefore, in most cases, the model results have errors due to interpolation (Please see **Comment 19a** for details.), and even increasing the resolution of models

does not seem to solve this problem (Liu et al., 2020). For the above reasons, we only give the total number of premature deaths in Chengdu instead of the precise distribution of premature deaths. And the uncertainties in the assessment of premature deaths are given in the revised manuscript. Please see lines 204-209 for details. Furthermore, we use percentage change to characterize the health risks associated with urbanization. Please see lines 456-477 for details.

$$\Delta M = Pop \cdot y_0 \left(1 - e^{-\beta(C-C_0)}\right)$$

As for significant digits in the citation, Liao et al. (2015) studied the impacts of urban expansion in the entire Yangtze River Delta with over 20 mega/large cities. They further analyzed the differences between daytime and nighttime, resulting in a large range of changes. Our results are averaged over the study period, and all numbers in this paper are accurate to one decimal place. Thanks for the reminder and reviewer #2's **Comment 1**. In the revised manuscript, we reorganize the language of the citation and review more studies on the impacts of urbanization in other regions. Please see lines 62-73 for details.

References

Li, Y., Zhao, X., Liao, Q., Tao, Y., and Bai, Y.: Specific differences and responses to reductions for premature mortality attributable to ambient $PM_{2.5}$ in China, Sci Total Environ, 742, 140643, https://doi.org/10.1016/j.scitotenv.2020.140643, 2020.

Liu, H., Liu, S., Xue, B., Lv, Z., Meng, Z., Yang, X., Xue, T., Yu, Q., and He, K.: Ground-level ozone pollution and its health impacts in China, Atmospheric Environment, 173, 223-230, https://doi.org/10.1016/j.atmosenv.2017.11.014, 2018.

Liu, T., Wang, C., Wang, Y., Huang, L., Li, J., Xie, F., Zhang, J., and Hu, J.: Impacts of model resolution on predictions of air quality and associated health exposure in Nanjing, China, Chemosphere, 249, 126515, https://doi.org/10.1016/j.chemosphere.2020.126515, 2020.

4. I found it rather interesting that the model simulations for both $PM_{2.5}$ and $O_3$ show that very high concentrations can also be observed for large distances downstream of Chengdu. Assuming that there are also large populations in the vicinity of Chengdu, the overall impact would thus be even larger. I understand that an analysis of the effects around the city would be clearly beyond the scope

of the paper, but I simply found this another interesting aspect.

*Response:* Thanks for the constructive comment. The phenomenon of downstream spread of air pollutants, also known as regional transport, has also been reported in many studies (Gong et al., 2020; Qiao et al., 2019). This is an important phenomenon and future research can be carried out in quantifying the contribution of Chengdu to air quality in adjacent areas, or vice versa.

Chengdu is situated on the Chengdu Plain that is widely known as "the Abundant Land" owing to its fertile soil, favorable climate and novel Dujiangyan Irrigation System. We agree that there are a large number of people in the vicinity of Chengdu, but the population density is very uneven since most of the area around Chengdu is mountainous. On the other hand, there are still many challenges in high-resolution simulation of mountainous areas. Therefore, it is difficult to obtain the precise distribution of health risks in mountainous areas, but there is no doubt that this is a topic worthy of continued effort.

References

Gong, C., Liao, H., Zhang, L., Yue, X., Dang, R., and Yang, Y.: Persistent ozone pollution episodes in North China exacerbated by regional transport, Environ Pollut, 265, 115056, 10.1016/j.envpol.2020.115056, 2020.

Qiao, X., Guo, H., Tang, Y., Wang, P., Deng, W., Zhao, X., Hu, J., Ying, Q., and Zhang, H.: Local and regional contributions to fine particulate matter in the 18 cities of Sichuan Basin, southwestern China, Atmos. Chem. Phys., 19, 5791-5803, 10.5194/acp-19-5791-2019, 2019.

**Specific comments**

1. Line 54: To which socioeconomic developments is the text referring?

*Response:* Thanks for the constructive comment. We reorganize the section of the introduction, and there is no such phrase in the revised manuscript.

2. Line 61: "the thermal contrast of the topography" is not entirely clear? Thermal circulations are caused by temperature gradients in the atmosphere.

*Response:* Thanks for the constructive comment. In the revised manuscript, in order to put our

research into context of the present study, we review the state of the art of mountain-air pollution research. The mountain-plain wind is caused by horizontal temperature differences between air over mountain massifs and the air over the surrounding plains. Please see lines 90-93 for details.

3. Line 66: I would suggest to be more precise here and specify in which ways the diffusion conditions are more complicated.

*Response:* Thanks for the constructive comment. Mountain wind systems, like mountain-plain wind, can often recirculate urban air pollutants and thus worsen air quality in mountainous cities. We emphasize this in the revised manuscript. Please see lines 93-96 for details.

4. Line 89: "Chengdu has the most complex terrain in the world". This is a strong statement. How is that defined and determined?

*Response:* Thanks for the constructive comment. We agree that this conclusion is arbitrary. In the revised manuscript, we have deleted this sentence.

5. Fig. 1: The figure shows $SO_2$ emissions, but their role is never actually mentioned in the paper. There are also high-emission areas in the southeast corner of the figure. Are these other large cities?

*Response:* Thanks for the constructive comment. We give the distribution of $SO_2$ emissions to illustrate that anthropogenic emissions in Chengdu are much larger than those in its surrounding areas. The high-emission areas in the southeast corner of the figure are from another megacity, Chongqing. While the straight-line distance between Chengdu and Chongqing is over 250 km. Thanks for the reminder and reviewer #2's **Comment 3**. We replace the original Figure 1b with the distribution of observed sites in Chengdu. Please see new Figure 1 for details.

6. Air quality and meteorological data: How many stations are there in Chengdu, i.e., the ones that are averaged for the analysis (line 129)? And how strongly do the locations differ?

*Response:* Thanks for the constructive comment. There are one sounding, one meteorological station and eight air quality stations in Chengdu. Please see Figure 1c for details. Since the air quality stations are mainly concentrated in the city center, we use the average of air pollutant concentrations

at all stations to represent air pollutant concentrations in Chengdu. In the revised Section 2.1 Air quality and meteorological data, we briefly describe these stations and the data processing. Please see lines 135-139 for details.

7. Line 137: At which height is the lowest model level? Is the boundary layer also properly resolved during the night?

*Response:* Thanks for the constructive comment. There are 32 vertical layers below the model top of 100 hPa and 12 vertical layers below 2 km to resolve the boundary layer processes. The size of the lowest vertical grid is ~25 m. We add this information in the revised manuscript. Please see lines 152-154 for details. Due to the lack of observations of the boundary layer, we are not entirely sure that the simulated boundary layer height is proper. However, in terms of the vertical distribution of soundings (Figure 4c, 4d, 5c and 5d) and pollutants (Figure 7a and 8a), the evolution of the simulated boundary layer is reasonable.

8. Line 175: The text says that β has units of $(\mu g\ m^{-3})^{-1}$, which makes sense to have a non-dimensional exponent in eq. 5, but Table 3 mentions units of %.

*Response:* Thanks for the constructive comment. *β* is the concentration-response function (CRF), which relates a unit change in air pollutant concentrations to a change in health endpoint incidence. It is identified by short- or long-term epidemiology studies. It is a ratio, and usually the estimated slope of the log-linear relation between concentration and mortality. In the revised manuscript, we assign a more precise meaning to *β*. Please see lines 191-194 for details.

9. Line 180 and Table 3: The references in the text and in the figure caption are not the same.

*Response:* Thanks for the constructive comment. *β* for $PM_{2.5}$ and MDA8 $O_3$ were from Chen et al. (2017) and Yin et al. (2017), respectively. *β* and $y_0$ included in Table 3 were summarized by Wang et al. (2021). In the revised manuscript, we unify the references in these two places. Please see lines 198-199 for details.

10. Lines 192 and 439: How are "pollution episodes" defined? Are these continuous periods, during

which the air quality standard is exceeded, i.e., they can have different lengths? Does "pollution" always mean exceeding the standards (e.g., line 232) in this paper?

*Response:* Thanks for the constructive comment. In China, the national ambient air quality standard for $PM_{2.5}$ is that daily average $PM_{2.5}$ concentrations cannot exceed 75 μg m$^{-3}$; the national ambient air quality standard for $O_3$ is that daily maximum 8-h average (MDA8) $O_3$ concentrations cannot exceed 160 μg m$^{-3}$. They have different lengths. Thus, Figure 2a shows the distribution of daily average $PM_{2.5}$ concentrations while Figure 2b shows the distribution of MDA8 $O_3$ concentrations. In the revised manuscript, we add a note on the definition of "pollution episodes". Please see lines 215-217 for details.

11. Line 213: What does "administrative division adjustment" mean?

*Response:* Thanks for the constructive comment. In May 2016, Jianyang urban district was placed under the jurisdiction of Chengdu, and thereby the population of Chengdu in 2016 increased by more than 1 million compared with 2015. Chengdu's population has grown rapidly both because of the influx of people and the consolidation of surrounding areas, which will further affect the premature deaths attributable to air pollutants. However, since our study is for Chengdu as a whole, this information seems to be redundant and has been removed from the manuscript.

12. Line 222: I am not sure how meaningful it is to calculate and discuss an increase or decrease between the first and last year if there is no real trend and the inter-annual variability is larger than the overall trend.

*Response:* Thanks for the constructive comment. $O_3$ concentrations monitored by the National Environmental Monitoring Center of China were reported in the unit of μg m$^{-3}$ under standard atmospheric conditions (273.15 K, 1 atm) before September 2018 and changed to 298.15 K conditions afterward. After revising this, we find that although the population of Chengdu was increasing from 2015 to 2021, the premature deaths due to $PM_{2.5}$ decreased while the premature deaths due to $O_3$ increased slightly. Please see lines 244-246 for $PM_{2.5}$ and lines 250-253 for $O_3$.

13. Table 4: Are these numbers calculated for individual days and then summed over the whole year?

Section 2.3 does not say for which time period the given equations are applicable. It is mentioned that equation 5 is applied to MDA8 O₃, but it does not say either over which time period the PM₂.₅ values have to be averaged.

*Response:* Thanks for the constructive comment. Yes, the numbers of premature death are first calculated for individual days and then are summed over the whole year/month. The MDA8 O₃ concentrations and daily average PM₂.₅ concentrations are used to calculate premature mortalities attributable to O₃ and PM₂.₅, respectively. Because the standard and $\beta$ for PM₂.₅ are based on the daily average concentrations of PM₂.₅ while those for O₃ are based on MDA8 O₃ concentrations. In the revised Section 2.3, we add the details of the calculations. Please see lines 202-204 for details.

14. Line 235: "cold westerly winds from the north" seems to be a contradiction. Are the winds from the west or from the north?

*Response:* Thanks for the constructive comment. We are sorry for this confusing sentence. The winds are from the west. In the revised manuscript, we have clarified this point. Please see lines 265-266 for details.

15. Line 239: Why do the humidity differences cause the inversion? Actually, it seems that the temperature decreases with height, i.e., there is no inversion, but only a stable layer.

*Response:* Thanks for the constructive comment. When Chengdu is controlled by the southerly warm air flow (purple arrow in Figure R4b), the inversion tends to appear around 700 hPa (purple rectangles in Figure R4c and d) under the effect of warm advection. Since the southerly airflow originating from the Bay of Bengal is rich in moisture, the humidity below the inversion is much higher than that above the inversion (The smaller the difference between the temperature and the dew point temperature, the higher the relative humidity.). Unlike January, there is no inversion in July (Figure 5c and 5d in the revised manuscript). In the revised manuscript, we analyze the meteorological conditions separately for January and July so that readers can distinguish the difference between the two backgrounds. Please see lines 261-290 for details.

[Figure]

**Figure R4. The weather charts at (a) 500 hPa and (b) 700 hPa for January 2017 based on the NCEP FNL reanalysis data. The purple pentacles show the location of Chengdu. The skew-T diagram at (c) 00:00 UTC and (d) 12:00 UTC in January 2017. The red and blue lines are the simulated air temperature and dew point temperature in Jan_Base simulation, while the red and blue points are the sounding temperature and dew point temperature. These results are monthly averages.**

16. Fig. 3: The 850-hPa level should be lower than the Tibetan Plateau. So what do the plotted fields over the Tibetan Plateau actually show? The areas above 850 hPa should best be masked in the plots. Also, are these monthly averages?

*Response:* Thanks for the constructive comment. For various purposes, such as driving numerical models, reanalysis data are often interpolated to different pressure layers. We agree that the 850-hPa layer is lower than the Tibetan Plateau. The areas above the selected pressure layer are masked in the revised Figure 4 and 5. Please see line 273 for Figure 4 and line 292 for Figure 5. The weather charts and the temperature profiles for January and July are monthly averages. We have added a note about this in the captions to Figure 4 and 5.

17. Fig. 4: (i) What do the black dots to the right of the subfigures (where one usually finds the wind arrows) mean? (ii) Are the profiles from the radiosoundings or also from the NCEP analysis as Fig. 3? Are they monthly means? (iii) Why are both 00 and 12 UTC shown if the differences are never discussed?

*Response:* Thanks for the constructive comment. (i) The black dots to the right of the skew-T plots usually represent the locations of the wind barbs in the base map of the skew-T plot (https://www.ncl.ucar.edu/Applications/skewt.shtml#ex2). However, the wind barbs are missing in this study because the data on the winds are incomplete and we are mainly concerned with the temperature profiles. (ii) The profiles are from the radiosoundings instead of the NCEP FNL analysis data. And they are monthly averages as Figure 4a and 4b. Thanks for the reminder. We have added a note about this in the captions of Figure 4 and 5. Please see lines 278-279 for Figure 4 and lines 296-297 for Figure 5. (iii) The radiosounding data are available at 00:00 and 12:00 UTC, corresponding to 08:00 and 20:00 LST (LST is UTC+8h) in Chengdu, which can show the diurnal difference in the vertical distribution of the atmosphere. The good agreement between the vertical profiles in the model and the sounding data indicates that our model can capture the diurnal variation of the boundary layer.

18. Line 269: A comparison of RH may not actually say much about the model performance in simulating humidity correctly because it combines the effects of humidity and temperature. It might be better to use, e.g., mixing ratio or specific humidity if the observations are available to convert the units.

*Response:* Thanks for the constructive comment. We often use relative humidity in weather reports out of habit. But we agree that relative humidity is not an accurate indicator of atmospheric water vapor because it largely depends on temperature. In the revised manuscript, we use dew point instead of relative humidity (the dew point is included in the surface meteorological data). Please see lines 312-314 for details. The dew point is the temperature to which air must be cooled to become saturated with water vapor. It provides a measure of the actual amount of water vapor in the air–a higher dew point means there is more moisture in the air.

19. Fig. 5

a. Are the measurements averages over all stations in the city as suggested in the previous section? If so is the model output equally averaged over all urban grid cells or over all grid cells closest to the stations?

*Response:* Thanks for the constructive comment. There are one sounding, one meteorological station and eight air quality stations in Chengdu (Figure 1c). Therefore, for the sounding and meteorological variables, the simulated values are the average of all grid cells closest to the station. We use the same method to first calculate the simulated $PM_{2.5}/O_3$ concentrations of the eight stations, and then average the simulated values as $PM_{2.5}/O_3$ concentrations for the entire city. We do not distinguish between urban and rural stations since these stations are basically located in the city center. We acknowledge that this approximation will affect the results of premature deaths because pollutant concentrations and population densities vary in different parts of the same city. We give more clarification in the revised Section 2 Data and methods, including the handling of data and the uncertainty about premature deaths. Please see lines 135-139 and lines 202-209 for details.

b. The figure and figure caption do not say which of the shown datasets (black dots or colored lines) is the model and which the observations. Based on a statement in the text that the model underestimates RH, I assume that the observations are the dots.

*Response:* Thanks for the constructive comment. Yes, the black dots are the observations and the colored lines are the model results. In the revised Figure 6, we add this in the figure caption. Please see lines 323-325 for details.

c. Wind speed, RH, and temperature seem to have a resolution of only 1 m s$^{-1}$, 5 or 10%, and 1ºC, respectively, which makes it really difficult to compare the diurnal cycles because, e.g., wind speed remains basically constant for hours. This will also affect the quantitative comparison (correlation coefficient). Is this the actual resolution of the data?

*Response:* Thanks for the constructive comment. We recheck the observation data as well as the code for processing these data. Temperature, dew point temperature, wind speed and $PM_{2.5}/O_3$ concentration in the observation data are measured hourly with resolutions of 1 °C, 1 °C, 1 m s$^{-1}$ and 1 μg m$^{-3}$. Therefore, meteorological variables may have the same value for several hours, especially for wind speed since the wind speed is small. This may reduce the correlation coefficient

between observations and simulations. Nevertheless, the correlation coefficients of all variables can still exceed the 99% significance level, indicating that our simulations are generally reasonable.

d. Why is wind direction not shown?

*Response:* Thanks for the constructive comment. In the revised Figure 6, we replace the wind speed in the original Figure 5 with the wind barbs to illustrate the wind direction. Northerly winds prevail in Chengdu in both January and July. Please see line 323 for details.

20. Line 292: "mountain-valley breezes" – Where can they be seen in the figure? Chengdu seems to be located in a northeasterly flow directed towards the mountains all day and night. How does that agree with diurnal circulations?

*Response:* Thanks for the constructive comment. After repeated discussions with the co-authors, we agree that this study is a classic example of regional mountain-plain circulation rather than local mountain-valley circulation. During daytime, the plain-to-mountain wind (plain wind) characterized by easterly and upslope flows (Figure R5c and d) draws $PM_{2.5}$-rich air mass from the polluted area to the eastern slope of the Tibetan Plateau. During nighttime, the circulation reverses, the mountain-to-plain wind (mountain wind) enhances $PM_{2.5}$ concentration over Chengdu (Figure R5a and b). Moreover, under the influence of the prevailing northeasterly wind (Figure 6), high-concentration $PM_{2.5}$ can be transported to downstream areas. These rules also apply to $O_3$ (Figure 8), which is the climatology of mountain areas. In the revised manuscript, we replace the mountain-valley breeze with the mountain-plain wind, and we explore the impact of mountain-plain wind on $PM_{2.5}$ and $O_3$ pollution in Chengdu. Please see lines 335-343 for $PM_{2.5}$ and lines 360-367 for $O_3$.

[Figure]

**Figure R5. Horizontal distributions of PM$_{2.5}$ with wind vectors at the lowest model level at (a) 2:00 and (b) 14:00 LST. East-west vertical cross sections of PM$_{2.5}$ with wind vectors at (c) 2:00 and (d) 14:00 LST. Purple pentacles show the location of Chengdu. Brown-shaded areas represent the terrain. These results are the monthly average based on Jan_Base simulation. The blue and magenta arrows highlight the mountain wind and plain wind, respectively.**

21. Line 294: It is very difficult to see in the horizontal cross sections whether there are actually converging flows, because the figures show the flow at the first model level (I assume so because the figure caption does not say), which means very large elevation differences. The shown wind field is about 2-3 km higher in the leftmost part of the figure than in the right part. Based on the figures, one could also argue that the high concentrations are simply located at the foot of the mountains, i.e., they are trapped by the terrain.

*Response:* Thanks for the constructive comment. Yes, horizontal distributions of PM$_{2.5}$, O$_3$ and wind

are the results at the lowest model level. We have added this to the figure caption in the revised manuscript. Please see lines 347-348 for PM$_{2.5}$ and lines 375-376 for O$_3$. The position of the convergent flows is shown by the green oval in Figure R5a, where the terrain is relatively flat (the green rectangle in Figure R5c). Driven by northeasterly wind and mountain wind, surface PM$_{2.5}$ converges to the west boundary of the Chengdu Plain and then will be transported to the eastern Tibetan Plateau edge and the northern Yunan-Guizhou Plateau edge areas, where it will be trapped by the terrain.

22. Line 298: Is this really a secondary circulation that is forced by the terrain? Could it not be the upper-level westerly flow that is shown in Fig. 3? This would also agree with the absence of this "secondary circulation" in July, when a weak-gradient situation prevails at upper levels.

*Response:* Thanks for the constructive comment. We are sorry for not giving a clear definition of the secondary circulation. The secondary circulation here refers to the mountain-plain wind (blue and magenta arrows in Figure R5c and d) rather than upper-level westerly flow. The daytime heating and nighttime cooling contrast between the mountain atmosphere over the outer slopes of the mountain massif and the free atmosphere over the surrounding plain produces the horizontal pressure differences that drive this wind system. In July, under the influence of the high-pressure systems, the horizontal pressure gradient is small so the mountain-plain wind is not as pronounced as in January. Nevertheless, it is still distinguishable (Figure 8). In the revised manuscript, we use the mountain and plain wind directly instead of vague words, i.e., the secondary circulation. Please see lines 335-343 for PM$_{2.5}$ and lines 360-367 for O$_3$.

**Technical comments, typos, etc.**

1. Several of the figure axis labels are very small and thus almost impossible to read, e.g., axes labels in Fig. 2 (in particular the days of the week, but also the months), Fig. 4., Figs. 6-11.

*Response:* Thanks for the constructive comment. In the revised manuscript, we have enhanced the quality of these figures with higher resolution and larger font size. Furthermore, we have redrawn other figures in this paper to make them clear.

2. For past events, the past tense should be used: e.g., "increased by 24.2%" (line 214). Similarly, lines 217, 232 and 438. There may be others.

*Response:* Thanks for the constructive comment. We carefully check the tense in this paper, using the past tense for past events and the present tense for present events.

3. Lines 25 and 30: It would be good to add that the mentioned changes in $PM_{2.5}$ concentrations are based on monthly means.

*Response:* Thanks for the constructive comment. In the revised manuscript, we add that changes in $PM_{2.5}$ and $O_3$ are based on monthly means. Please see lines 28-31 and lines 33-35 for details.

4. Line 26: MDA8 is not defined

*Response:* Thanks for the constructive comment. We add the definition of MDA8 to the revised abstract. Please see line 29 for details.

5. Line 55: "increases in anthropogenic emissions"

*Response:* Thanks for the constructive comment. We have unified the singular and plural forms here as well as in the rest of this paper.

6. Line 60: The reference list contains only Qian et al. (2022).

*Response:* Thanks for the constructive comment. We correct this typo to Qian et al., 2022 in the revised manuscript. Please see line 75 for details.

7. Line 67: "notable pollution episodes"

*Response:* Thanks for the constructive comment. We reorganize the section of the introduction, and there is no such phrase in the revised manuscript.

8. Line 67: Chow et al. (2013) is not included in the reference list.

*Response:* Thanks for the constructive comment. We recheck and update all references in this paper, including the addition of the book titled Mountain weather research and forecasting edited by Chow

et al. (2013). Please see line 90 and lines 749-751 for details.

9. Line 71: I don't understand the meaning of the sentence "Although the principles …"

*Response:* Thanks for the constructive comment. We reorganize the section of the introduction, and there is no such sentence in the revised manuscript.

10. Line 91: It would be really helpful to label these terrain features in Fig. 1a?

*Response:* Thanks for the constructive comment. We take your suggestion and mark the names of these mountains in the new Figure 1a. Please see lines 122-125 for details.

11. Fig. 1: It would be helpful to indicate the location of Chengdu in subfigure (b). I assume it corresponds to the red location in the center with high emission values?

*Response:* Thanks for the constructive comment. We remove the original Figure 1b since the role of $SO_2$ emission is never actually mentioned in the paper. Instead, we give the boundary of Chengdu and the distribution of monitoring sites in Chengdu. We mark the location of Chengdu in Figure 1a and 1b. Please see lines 122-125 for details.

12. Line 115: "Data and methods" might be a more common section title.

*Response:* Thanks for the constructive comment. The title of Section 2 has been changed from "Materials and methods" to "Data and methods". Please see line 127 for details.

13. Line 200: "The high temperature and strong sunlight contribute to the elevated …"

*Response:* Thanks for the constructive comment. In the new manuscript, the sentence "Elevated $O_3$ concentrations in warm months are contributed to the high temperature and strong sunlight." is revised to "The high temperature and strong sunlight contribute to the elevated $O_3$ concentrations in warm months". Please see lines 229-230 for details.

14. Line 189: "two crucial air pollutants that account for air pollution". This is a somewhat awkward sentence. "that account for air pollution" can easily be removed.

*Response:* Thanks for the constructive comment. In the new manuscript, the awkward sentence "$PM_{2.5}$ and $O_3$ are two crucial air pollutants that account for air pollution." has been removed.

15. Line 237: "to the west of Chengdu"

*Response:* Thanks for this comment and **Specific comments 16**. Since the 850-hPa level is much lower than the Tibetan Plateau, the Southwest Vortex that appears to the west of Chengdu is masked out on the weather charts. Thus, the corresponding sentence does not exist in the revised manuscript.

16. Line 240: "cold air"

*Response:* Thanks for the constructive comment. The phrase "cold air" is revised to "blocking of air", and the entire sentence is "The blocking of air and the temperature inversion were two important reasons for frequent $PM_{2.5}$ pollution episodes during this period.". Please see lines 269-271 for details.

17. Line 314: "and mixing" (remove "well")

*Response:* Thanks for the constructive comment. The word "well" has been removed in the revised manuscript. Please see lines 359-360 for details.

18. Line 360: "can increase" suggests that this is a maximum value. Or is the text also referring to a monthly average as for $PM_{2.5}$? It would actually help to always add "monthly mean" when talking about monthly mean values (e.g., line 404) to avoid misunderstandings.

*Response:* Thanks for the constructive comment. In the revised manuscript, we take your suggestion and always add "monthly mean" when we talk about monthly mean values, including in the abstract, text and figure captions.

19. Fig. 10: Is there a reason why the color scale is not adjusted to show the whole range? Based on the color bar it looks like the highest values are around 10 $\mu g\ m^{-3}$, whereas the text actually mentions a monthly average of 26.6 $\mu g\ m^{-3}$.

*Response:* Thanks for the constructive comment. Figures 9-12 use the same color bar to visualize

the different effects of urban land use and anthropogenic emission on $PM_{2.5}$ and $O_3$. In the revised manuscript, we adjust the color scale in Figures 9-12 so that they can show the whole range of pollutant concentrations. Please see lines 395, 416, 433 and 451 for Figure 9, 10, 11 and 12, respectively.

20. Line 407: "assess" instead of "access"

*Response:* Thanks for the constructive comment. In the revised manuscript, we have corrected this typo. Please see line 461 for details.

21. Line 545: This reference is not cited in the manuscript.

*Response:* Thanks for this comment and **Technical comment 8**. This reference is incorrectly cited in the text as Chow et al. (2013) because the book was edited by Prof. Chow. We have corrected this typo in both the text and references. Please see line 90 and lines 749-751 for details.

**Anonymous Referee #2:**

**General comments**

Accelerated urbanization across the globe has resulted in considerable changes in land surface parameters and subsequently affect meteorological conditions and air pollutant levels. In this study, Zhan et al. reveal the changes in criteria air pollutants between 2015 and 2021 and probe the environmental consequence of rapid urbanization over a typical megacity situated in southwestern China, Chengdu. They also quantify the premature mortalities attributed to exposure to ozone and $PM_{2.5}$. This paper is well-written and presents results that would be interesting to the air quality modeling community from a practical perspective. I have several concerns that the authors should consider when revising the manuscript, as listed below. I recommend publication after the following comments are adequately addressed.

*Response:* We would like to thank the referee for the valuable and affirmative comments on our manuscript. We have carefully revised our manuscript based on the following comments.

**Specific comments**

1. The literature review could be better. The authors only provide an example of urbanization impacts focused on the YRD by Liao et al. (2015), while extensive studies have been focusing on identifying the effects of urbanization on the regional meteorological phenomena and air quality (including the Beijing-Tianjin-Hebei area and the Pearl River Delta). Furthermore, recent studies have widely acknowledged the critical role of urbanization in altering air quality in Chengdu [Wang et al., 2021, 2022a]. Thus, these studies should be discussed and properly cited.

*Response:* Thanks for the constructive comment. We take your suggestion and deeply reorganize the introduction, in which we add a review of research on the effects of urbanization on regional meteorology and air quality in other regions, such as the Beijing-Tianjin-Hebei region, the Pearl River Delta region and the Sichuan Basin. Please see lines 62-73 in the revised manuscript for details. Thanks for the recommended literatures. We have learned a lot from these literatures, all the recommended literatures are cited in the revised manuscript.

2. Line 34-36: This sentence is a bit vague and I genuinely don't understand this sentence - please rephrase.

*Response:* Thanks for the constructive comment. The original sentence "This reminds us that the development of cities is also important for the urban air quality apart from the emissions reduction" is revised as "This reminds us that, in addition to regulating anthropogenic emissions, urban planning is also important for the urban air quality, especially for secondary pollutions like $O_3$." in the new manuscript. Please see lines 40-41 for details.

3. Figure 1: It seems that the shapefile used by the authors (NCL default shapefile) is wrong. Please check. Also, please clarify the source of $SO_2$ emission for subplot (b). Moreover, it would be valuable to provide the boundary of Chengdu city in Figure 1 for making it clear.

*Response:* Thanks for the constructive comment. We have revised the NCL default shapefile based on the map database provided by Dr. Yongjie Huang (https://github.com/huangynj/NCL-Chinamap). Thanks for the reminder and reviewer #1's **specific comment 5**. We remove the original Figure 1b since the role of $SO_2$ emission is never actually mentioned in the paper. Instead, we give the

boundary of Chengdu and the distribution of monitoring sites in Chengdu. Please see the new Figure 1 in line 123.

4. Line 190: The criterion for PM$_{2.5}$ and MDA8 ozone is a bit vague. I believe that it is annual PM$_{2.5}$ concentrations less than 75μg/m$^3$ and MDA8 ozone less than 160μg/m$^3$. Please clarify the time period for these metrics.

*Response:* Thanks for the constructive comment. In China, the national ambient air quality standard for PM$_{2.5}$ is that daily average PM$_{2.5}$ concentrations cannot exceed 75 μg m$^{-3}$; the national ambient air quality standard for O$_3$ is that MDA8 O$_3$ concentrations cannot exceed 160 μg m$^{-3}$. The definition of PM$_{2.5}$/O$_3$ pollution is based on Chinese national standards. We have clarified these metrics in the revised manuscript. Please see lines 215-217 for details.

5. Line 193-196: The authors apply the annual mean MDA8 ozone concentrations for illustrating the variations of ozone across Chengdu over time. However, annual mean MDA8 ozone is not a meaningful metric as wintertime low MDA8 ozone would pull low ozone levels. In general, it is recommended for using the warm season (April-September) MDA8 ozone average (see Wang et al., (2022b)) or the 90th percentile of MDA8 ozone (which is based on Chinese NAAQS GB3095-2012).

*Response:* Thanks for the constructive comment. We agree that the annual mean MDA8 O$_3$ concentration is not a meaningful metric for the interannual variation of O$_3$. Since the annual metric for O$_3$ in China is the 90$^{th}$ percentile of MDA8 O$_3$ (GB3095-2012; HJ663-2013), we adopt the 90$^{th}$ percentile of MDA8 O$_3$ concentrations in the revised manuscript. Please see lines 221-225 for details.

6. It would be better to use "heat maps of (a) daily average PM$_{2.5}$ and (b) MDA8 O$_3$ concentrations" rather than "distribution" for the caption of Figure 2.

*Response:* Thanks for the constructive comment. The new caption of Figure 2 has been revised to "Figure 2. Heat map of (a) daily PM$_{2.5}$ and (b) MDA8 O$_3$ concentrations in Chengdu from 2015 to 2021." Please see lines 233-234 for details.

7. Line 219-220: "the premature mortalities due to O$_3$ fluctuate." This is an incomplete sentence.

Please check.

*Response:* Thanks for the constructive comment. In the revised manuscript, this incomplete sentence is removed. Instead, we directly give the number of premature deaths attributable to $O_3$. Please see lines 247-250 for details.

8. Line 221: "annual average" might be "7-year average". Please check.

*Response:* Thanks for the constructive comment. Yes, it is "7-year annual average" instead of "annual average". In the revised manuscript, we have corrected this typo. Please see lines 22, 240-243, 250 and 492-493 for details.

9. Line 275-278: Is the WRF-Chem model performance comparable with prior studies over Chengdu (or Sichuan Basin)? It would be valuable to briefly compare the model performance with previous studies (Yang et al., 2021; Wu et al., 2022) for demonstrating the robustness of model results.

*Response:* Thanks for the constructive comment. In the previous version, we do not compare the WRF-Chem model performance with prior studies over Chengdu. Thanks for the recommended literatures. We add a brief comparison of our model results with those from prior studies in the revised manuscript, which further suggests that our simulations are reasonable. Please see lines 305-311 and lines 314-318 for details.

10. Line 369-370: This sentence is a bit vague and I genuinely don't understand this sentence - please rephrase.

*Response:* Thanks for the constructive comment. We apologize for this confusing sentence here. In the revised manuscript, a new sentence "Rising anthropogenic emissions of air pollutants and their precursors can significantly increase ambient air pollution." is adopted. Please see lines 423-425 for details.

11. Line 385-395: The authors attribute the ozone changes in Chengdu to the Ozone-NOx-VOCs regime but do not provide any details about the formation regime. A comprehensive discussion on

the underlying mechanism of the VOCs-limited ozone regime in urban Chengdu is needed (Wang et al., 2022a).

*Response:* Thanks for the constructive comment. Yes, we attribute only a slight increase in $O_3$ concentrations from anthropogenic emissions to the non-linear sensitivity of $O_3$ and its precursor (VOCs and $NO_x$). Thanks to your literature and references therein, in the revised manuscript, we add a discussion on $O_3$ formation regime in Chengdu. From 2013 to 2020, metropolitan Chengdu remains VOCs-limited regime, and the effect of reducing $NO_x$ emissions may be partially offset by changes in VOCs. Please see lines 440-446 for details.

**References**

[1] Wang, H., et al. (2021). Impact of different urban canopy models on air quality simulation in Chengdu, southwestern China. Atmospheric Environment, 267, 118775. https://doi.org/10.1016/j.atmosenv.2021.118775

[2] Wang, H., et al. (2022a). Impact of Urbanization on Meteorology and Air Quality in Chengdu, a Basin City of Southwestern China. Frontiers in Ecology and Evolution, 10, 845801. https://doi.org/10.3389/fevo.2022.845801

[3] Wang, Y., et al. (2022b). Long-term trends of ozone and precursors from 2013 to 2020 in a megacity (Chengdu), China: Evidence of changing emissions and chemistry. Atmospheric Research, 106309. https://doi.org/10.1016/j.atmosres.2022.106309

[4] Wu, K., et al. (2022). Drivers of 2013–2020 ozone trends in the Sichuan Basin, China: Impacts of meteorology and precursor emission changes. Environmental Pollution, 300, 118914. https://doi.org/10.1016/j.envpol.2022.118914

Thanks for the recommended literatures. We have learned a lot from these literatures, all the recommended literatures are cited in the revised manuscript. Please see lines 719-721, 722-724, 729-732 and 735-738 of the references.

---

## Referee Report (RR1)

I appreciate the work that the authors have put into revising the manuscript and responding to my previous comments. I think the paper has greatly improved as a result of the additional figures that show now the previously only hinted at connections between the boundary-layer height or the surface temperature and the changes in pollutant concentrations. I think it is also great that the authors decided to show only two time steps in the main figures and instead increased the size of the subfigures.

When reading through the paper, I noticed a few additional, but very minor, points, which the authors may want to address before final publication. The line numbers in my comments refer to the revised version without track changes.

**Specific comments**

- 1) You mentioned in response to one of my previous comments that you cut the anthropogenic emission data from the MEIC inventory by about 20%. I think this is important information that should also be included in the paper.
- 2) line 24: 95% CI is not defined in the abstract.
- 3) Line 26ff: "The results show that urban land use led to an increase ... compared to cropland, which was conducive ..."
- 4) line 113: I would say you are investigating "the impacts of urbanization on air pollutant concentrations" rather than "on air pollutants" themselves. Similar on line 116.
- 5) line 135: You may want to say "the urban hourly pollutant concentrations reported in this paper" to make it clear that this is not some standard parameter.
- 6) Line 141: "10-m wind direction"
- 7) Table 3: I am still confused about the units of β. In Table 3, the units are given as %, but the text says that it is "the percent change of mortality per 10 µg m-3 increase ...", i.e., the units would be "% (10 µg m-3)-1.
- 8) Fig 3 caption: Please add the subfigure labels to the caption: "attributable to (a) PM2.5 and (b) O3"
- 9) Line 266: Do you mean "dispersion" instead of "elimination"?
- 10) Line 268: I guess you mean "inversion layer" instead of "inverse layer"? The temperature observations, however, do not show a temperature inversion if I connect the dots, i.e., the temperature is not increasing with height. This may be a result of the really coarse vertical resolution with only 6 data points. When I looked up the soundings at weather.uwya.edu, I noticed that the soundings contain actually more data points. Why did you not use the full resolution? If it is because of the computation of the monthly mean, you could first interpolate the soundings to a common vertical grid and then average. I think the comparison of the model with the soundings would really profit from a better vertical resolution. Coming back to the term "inversion" layer, which really means an increase in temperature with height and not just a stable layer, even the mean model profile shows only a more or less isothermal layer, but not a proper inversion.
- 11) Fig. 4: I would suggest to remove the black dots if they have no meaning, because they cause only confusion otherwise.
- 12) Line 285: Do you mean Figure 6b instead of 5b?
- 13) Figure 6: I really appreciate the authors' attempt to include wind direction in the figure following my previous comment. However, I am afraid the solution with wind barbs is sub-optimal as it is almost impossible to see the individual barbs. It might be better to simple plot time series of both wind speed and direction.
- 14) Section 3.3.2 and Fig. 6: Just a comment: One point that may also affect the comparison between the model and the observations (negatively) is the height difference. The model results are from the first model level, whereas the temperature observations are at 2 m above ground.

- 15) Lines 332 and 333: I would suggest to either say "boundary layer depth" instead of "height" or add "above ground" to the heights.
- 16) Lines 339ff: I don't understand the argument. The northeasterly flow would transport PM2.5 away from the slope, i.e., down the eastern slope (and not lift the air up along the slope), which then leads to the described large downstream spread.
- 17) Figs. 7 and 8: Subfigure (a) does not have an x axis label (Time).
- 18) Figs. 7 and 8 caption: I think it would be helpful to add the line types to the description of (a), e.g., "... cross sections of PM2.5 (color shading), potential temperature (purple contour lines), and boundary layer height (thick black contour line) ...". Also, how is the boundary layer height determined? Is it the output from the PBL scheme or did you determine it directly from the model 3D fields?
- 19) Line 366: Do you mean "... carry O3-rich air eastward"?
- 20) Line 404: I would suggest to add "compared to cropland" after "induced by urban land use".
- 21) Line 411: Do you mean "... with the monthly average value increasing by 5.4 ..."?
- 22) Line 464: "with the existence of Chengdu" I assume you are referring to the urban land use compared to cropland? The text is a bit unclear because anthropogenic emissions are also related to the existence of Chengdu.
- 23) Fig. 13: I find it somewhat confusing that the legend entries contain the same set of symbols, but with different labels. You could maybe use different colors for the left and right side of the figure.
- 24) Fig. 13 caption: Please explain in the caption what the dots (average?) and the whiskers (95% CI?) are.
- 25) Line 506: Are you again referring to monthly averages, i.e., "monthly averaged surface PM2.5 concentrations"?

**Typos**

- 1) Line 22: "the 7-year annual averages"
- 2) Throughout the document, ranges are given with a ~ instead of a (e.g., line 24: 6542~11726)
- 3) Line 28: "could decrease" You observed this decrease in your simulations, so you don't need to say "could", simply say "decreased". Similar on lines 30 and 34.
- 4) Line 93: "During daytime" instead of "During daydurtime"
- 5) Line 153: Maybe better say "The height of the lowest model level" instead of "The size of the lowest vertical grid"
- 6) Line 220: "and for  $O_3$  it is" instead of "and it for  $O_3$  is"
- 7) Line 224: "PM2.5 pollution has improved ... O3 pollution has not" or "PM2.5 pollution improved ... O3 pollution did not"
- 8) Line 226: "that is" instead of "that was"
- 9) Fig 3 caption: "ANAC" instead of "ANA".
- 10) Line 301: "troposphere" instead of "tropospheric atmosphere"
- 11) Line 308: Remove (0.44) and (0.77) from the sentence, since these numbers are already contained in the main sentence.
- 12) Line 355: I guess you mean "downward" instead of "downstream".
- 13) Lines 465, 498, and 501: "While" is usually used to start a sub-clause, but not a main clause without a sub-clause. You probably mean something like "however" or "on the other hand", e.g., "On the other hand, anthropogenic emissions ...".

---

## Author Response (AR2)

**Response to Reviewer**

**Dear referee,**

We would like to thank you very much for your professional comments on our manuscript "Impacts of urbanization on air quality and related health risks in a city with complex terrain" (egusphere-2022-486). According to these comments, we have carefully revised the manuscript again. Here are point-by-point responses (in blue color), and the changes are reflected in the revised manuscript (in red color). The line numbers in the authors' responses are obtained from the revised manuscript, in which all the revisions have been accepted.

I appreciate the work that the authors have put into revising the manuscript and responding to my previous comments. I think the paper has greatly improved as a result of the additional figures that show now the previously only hinted at connections between the boundary-layer height or the surface temperature and the changes in pollutant concentrations. I think it is also great that the authors decided to show only two time steps in the main figures and instead increased the size of the subfigures.

*Response:* We are grateful for the positive evaluation and constructive comments on our manuscript. When reading through the paper, I noticed a few additional, but very minor, points, which the authors may want to address before final publication. The line numbers in my comments refer to the revised version without track changes.

We appreciate your time and effort to improve the quality of our manuscript. We have addressed each comment in detail and the responses are listed below.

**Specific comments**

 You mentioned in response to one of my previous comments that you cut the anthropogenic emission data from the MEIC inventory by about 20%. I think this is important information that should also be included in the paper.

*Response:* We agree with you. In the Section 2.2 WRF-Chem model and experimental designs, we add a note as follows: "*It should be noted that we empirically cut the PM*2.5 *emissions by about 20% to avoid overestimation of PM*2.5 *in the model.*". Please see lines 159–160 for details.

2) line 24: 95% CI is not defined in the abstract.

*Response:* Thanks for the constructive comment. We have added the definition of 95% CI to the revised abstract. Please see line 24 for details.

 Line 26ff: "The results show that urban land use led to an increase ... compared to cropland, which was conducive ..."

*Response:* Thanks for the constructive comment. This sentence is revised as "*The results show that urban land use led to an increase in air temperature and the boundary layer height compared to cropland*, which was conducive to the diffusion of  $PM_{2.5}$ .". Please see lines 26–28 in the revised abstract.

 line 113: I would say you are investigating "the impacts of urbanization on air pollutant concentrations" rather than "on air pollutants" themselves. Similar on line 116.

*Response:* We agree with you and "air pollutants" has been replaced by "air pollutant concentrations" in the revised manuscript. The revised sentence is "*In this study, we investigate the impacts of urbanization on air pollutant concentrations and the corresponding health risks in Chengdu.*" (lines 113–114). Similar on lines 116–117.

 line 135: You may want to say "the urban hourly pollutant concentrations reported in this paper" to make it clear that this is not some standard parameter.

*Response:* Thanks for the clarification. The revised sentence is "*There are eight air quality stations* throughout Chengdu, and the urban hourly pollutant concentrations **reported in this paper** are the average results of measurements at all monitoring sites.". Please see lines 135–137 for details.

**6) Line 141: "10-m wind direction"**

*Response:* We are sorry for this mistake. "10-m direction" has been corrected to "10-m wind direction" in line 141 of the revised manuscript.

7) Table 3: I am still confused about the units of  $\beta$ . In Table 3, the units are given as %, but the text says that it is "the percent change of mortality per 10 µg m-3 increase ...", i.e., the units

would be "%  $(10 \ \mu g \ m^{-3})^{-1}$ .

*Response:* Thanks for the constructive comment. We delete the unit of  $\beta$  from Table 3. Instead, we directly give the meaning of  $\beta$  in the note to Table 3, that is, " $\beta$  *is expressed as the percentage increase (posterior mean and 95% confidence intervals) in daily mortality associated with a 10 µg m-3 increase in daily PM2.5/MDA8 O3 concentrations." (lines 214–215). We hope this will help readers understand \beta.*

Fig 3 caption: Please add the subfigure labels to the caption: "attributable to (a) PM2.5 and (b) O3"

*Response:* Thanks for the constructive comment. The revised caption of Figure 3 is "*Premature* mortality from ANAC, CVD, RD and COPD attributable to (a) PM2.5 and (b) O3 in Chengdu from 2015 to 2021. The dots represent the mean estimate, and the whiskers represent 95% confidence intervals.". Please see lines 261–263 for details.

9) Line 266: Do you mean "dispersion" instead of "elimination"?

*Response:* Yes, it is "dispersion" instead of "elimination". The complete sentence is "*However, the westerly winds were blocked by the Tibetan Plateau and thereby the* **dispersion** of *PM*2.5 was limited." Please see lines 271–272 for details.

10) Line 268: I guess you mean "inversion layer" instead of "inverse layer"? The temperature observations, however, do not show a temperature inversion if I connect the dots, i.e., the temperature is not increasing with height. This may be a result of the really coarse vertical resolution with only 6 data points. When I looked up the soundings at weather.uwya.edu, I noticed that the soundings contain actually more data points. Why did you not use the full resolution? If it is because of the computation of the monthly mean, you could first interpolate the soundings to a common vertical grid and then average. I think the comparison of the model with the soundings would really profit from a better vertical resolution. Coming back to the term "inversion" layer, which really means an increase in temperature with height and not just a stable layer, even the mean model profile shows only a more or less isothermal layer, but not a proper inversion.

*Response:* The pressure levels of the sounding data vary from day to day except at a few specific pressure levels (925 hPa, 850 hPa, 700 hPa, 500hPa, 400 hPa, 300 hPa and 250 hPa). Therefore, we have previously only given monthly mean results for these specific pressure layers. Thanks for your suggestion. In the revised manuscript, we supplementarily calculate the monthly mean between these specific pressure layers and finally obtain data for 14 pressure layers. These 14 pressure layers are ~950 hPa, 925 hPa, ~885 hPa, 850 hPa, ~765 hPa, 700 hPa, ~620 hPa, 500 hPa, ~450 hPa, 400 hPa, ~350 hPa, 300 hPa, ~275 hPa and 250 hPa. Since the vertical resolution increases, we show the observed profiles with dashed lines instead of dots in revised Figure 4c and 4d. As shown in the revised Figure 4c and 4d (line 279), the temperature inversion layer near 700 hPa is indeed not obvious. Thus, we agree it should be "stable layer" instead of "inversion layer" here. We have corrected this point in the revised manuscript. Please see lines 273–275 for details.

 Fig. 4: I would suggest to remove the black dots if they have no meaning, because they cause only confusion otherwise.

*Response:* Thanks for the constructive comment. The black dots have been removed from the revised Figure 4c, 4d, 5c and 5d. Please see line 279 for Figure 4c and 4d, line 297 for Figure 5c and 5d.

12) Line 285: Do you mean Figure 6b instead of 5b?

*Response:* We are sorry for this mistake. "Figure 5b" has been corrected to "Figure 6b" in line 291 of the revised manuscript.

13) Figure 6: I really appreciate the authors' attempt to include wind direction in the figure following my previous comment. However, I am afraid the solution with wind barbs is suboptimal as it is almost impossible to see the individual barbs. It might be better to simple plot time series of both wind speed and direction.

*Response:* Thanks for the constructive comment. We remove the wind barbs and present the time series of 10-m wind speed ( $WS_{10}$ ) and 10-m wind direction ( $WD_{10}$ ) in the revised Figure 6. Since  $WD_{10}$  does not change continuously, we use cyan dots to represent the simulated  $WD_{10}$  for aesthetics. As shown in Figure 6 (line 331), the frequency of calm wind is high due to the starting speed of the

anemometer (typically 0.5-1 m/s). In this case, the simulated wind speed must be greater than the observed one, resulting in an overestimation of the simulated WS10. Except for the case of calm wind, our model can generally capture the shift in WD10 during the study period. Therefore, the modeling results of 10-m wind are reasonable and acceptable.

- 14) Section 3.3.2 and Fig. 6: Just a comment: One point that may also affect the comparison between the model and the observations (negatively) is the height difference. The model results are from the first model level, whereas the temperature observations are at 2 m above ground. *Response:* We agree that observations and simulations at different altitudes affect the results of their comparisons. 2-m air temperature (T2) and O3 concentrations are state variables and can directly be obtained from the WRF output files. However, 2-m dew point temperature (TD2) and 10-m wind are diagnostic variables, which are recommended to be calculated by function wrf\_user\_getvar in NCL (https://www2.mmm.ucar.edu/wrf/OnLineTutorial/Graphics/NCL/NCL\_functions.php). This method will inevitably have some errors.
- 15) Lines 332 and 333: I would suggest to either say "boundary layer depth" instead of "height" or add "above ground" to the heights.

*Response:* Thanks for the constructive comment. We have added "above ground" to the heights. The revised sentences become "*the boundary layer height was only* ~320 *m* **above ground**." (lines 340–341) and "*The daytime atmospheric boundary layer, also known as the convective boundary layer, could develop to* ~1300 *m* **above ground**." (lines 342–343).

16) Lines 339ff: I don't understand the argument. The northeasterly flow would transport PM2.5 away from the slope, i.e., down the eastern slope (and not lift the air up along the slope), which then leads to the described large downstream spread.

*Response:* Thanks for the constructive comment. You are right. The convergence of westerly mountain wind (blue arrow in Figure R1) and northeasterly wind (magenta arrow in Figure R1) was conducive to the formation of  $PM_{2.5}$  pollution belt (green circle in Figure R1) and its spread to the downstream of Chengdu. In the revised manuscript, we have reformulated these results. Please see lines 346–349 for details.

Figure R1. Horizontal distribution of  $PM_{2.5}$  with wind vectors at the lowest model level at 2:00 LST. Purple pentacle shows the location of Chengdu.

17) Figs. 7 and 8: Subfigure (a) does not have an x axis label (Time).*Response:* Thanks for the constructive comment. We have added the x axis label (Time) to the revised Figure 7a and 8a. Please see lines 354 and 382 for details.

18) Figs. 7 and 8 caption: I think it would be helpful to add the line types to the description of (a), e.g., "... cross sections of PM2.5 (color shading), potential temperature (purple contour lines), and boundary layer height (thick black contour line) ...". Also, how is the boundary layer height determined? Is it the output from the PBL scheme or did you determine it directly from the model 3D fields?

*Response:* Thanks for the constructive comment. We accept the suggestion and have added the line types to the description of Figure 7a and 8a. The new captions of Figure 7a and 8a are "*Figure 7*. (a) Temporal-vertical cross sections of  $PM_{2.5}$  (color shading), potential temperature (purple contour lines) and boundary layer height (thick black contour line) at Chengdu." (lines 355–356) and "*Figure 8. (a) Temporal-vertical cross sections of O3 (color shading)*, potential temperature

*(purple contour lines)* and boundary layer height *(thick black contour lines)* at Chengdu." (lines 383–384), respectively.

In this study, planetary boundary layer physics options (bl\_pbl\_physics) used the Mellor-Yamada-Janjic Scheme (MYJ). The MYJ scheme, based on the turbulence kinetic energy (TKE) budget equation, determined the boundary layer height (PBLH) as where the TKE decreases to a prescribed small value ( $0.2 \text{ m}^2 \text{ s}^{-2}$ ). The prognostic equation for TKE is solved by using diagnostic estimation of potential temperature, water vapor variance, and covariances (Tyagi et al., 2018). PBLH is a state variable and is included in WRF output files.

**Reference**

- Tyagi, B., Magliulo, V., Finardi, S., Gasbarra, D., Carlucci, P., Toscano, P., Zaldei, A., Riccio, A., Calori, G., and D'Allura, A.: Performance analysis of planetary boundary layer parameterization schemes in WRF modeling set up over southern Italy, Atmosphere, 9, 272, 2018.
- 19) Line 366: Do you mean "... carry O3-rich air eastward"?

*Response:* Yes. The revised sentence is "*The nighttime mountain wind could carry rich-O*3 *air eastward and enhanced O*3 *concentrations aloft over the eastern slope of the Tibetan Plateau*.". Please see lines 375-376 for details.

20) Line 404: I would suggest to add "compared to cropland" after "induced by urban land use". *Response:* According to the suggestion, the revised sentence is "*Due to the increase in upward air movement and boundary layer height induced by urban land use compared to cropland, like PM*2.5, *NOx concentrations also decreased near the surface.*". Please see lines 412–414 for details.

21) Line 411: Do you mean "... with the monthly average value increasing by 5.4 ..."? *Response:* Yes. The revised sentence becomes " $O_3$  concentrations would also increase, with the monthly average value increasing by 5.4 µg m-3 (4.5%) at 14:00 LST.". Please see lines 420–421 for details.

22) Line 464: "with the existence of Chengdu" – I assume you are referring to the urban land use compared to cropland? The text is a bit unclear because anthropogenic emissions are also related to the existence of Chengdu.

*Response:* Yes, we are referring to the urban land use compared to cropland here. We are sorry for this unclear sentence and it has been revised to "the premature mortalities from ANAC, CVD, RD and COPD due to PM2.5 decreased by 171 (95%CI: 129–200, or about 6.9%), 45 (95%CI: 34–53, or about 6.7%), 22 (95%CI: 16–27, or about 6.5%) and 23 (95%CI: 17–26, or about 6.2%) in January 2017 when Chengdu area was urban land use rather than cropland.". Please see lines 472–475 for details.

23) Fig. 13: I find it somewhat confusing that the legend entries contain the same set of symbols, but with different labels. You could maybe use different colors for the left and right side of the figure.

*Response:* Thanks for the constructive comment. We accept the suggestion and use different colors for the left and right side of the revised Figure 13 to show the differences in premature mortality attributable to  $PM_{2.5}$  (left of the dotted line) and  $O_3$  (right of the dotted line). Please see line 490 for details.

24) Fig. 13 caption: Please explain in the caption what the dots (average?) and the whiskers (95% CI?) are.

*Response:* Thanks for the constructive comment. We have added an explanation of the dots (mean estimate) and the whiskers (95% confidence intervals) to the caption of Figure 13 as well as Figure 3. Please see lines 493–494 for Figure 13, and lines 262–263 for Figure 3.

25) Line 506: Are you again referring to monthly averages, i.e., "monthly averaged surface PM2.5 concentrations"?

*Response:* Yes. Thanks for the constructive comments. We have added "monthly averaged" before "surface PM2.5 concentrations" or "MDA8 O3 concentrations" in the revised conclusions. Please see lines 518 and 521–522 for details.

**Typos**

1) Line 22: "the 7-year annual averages"

*Response:* We are sorry for this mistake. "the 7-year annual average" has been corrected to "the 7-year annual averages" in line 22 of the revised manuscript.

2) Throughout the document, ranges are given with a ~ instead of a – (e.g., line 24: 6542~11726) *Response:* We accept your suggestion. In the revised manuscript, the symbol ~ is replaced by – throughout the document.

 Line 28: "could decrease" – You observed this decrease in your simulations, so you don't need to say "could", simply say "decreased". Similar on lines 30 and 34.

*Response:* Thanks for the constructive comment. Throughout the document, including the abstract and the conclusions, the modal auxiliary verbs before specific results are omitted. Please see lines 29, 30 and 34 for examples.

4) Line 93: "During daytime" instead of "During daydurtime"

*Response:* We have corrected this typo to "During daytime" in the revised manuscript. Please see line 93 for details.

 Line 153: Maybe better say "The height of the lowest model level" instead of "The size of the lowest vertical grid"

*Response:* We agree with you and this sentence is clarified as follows: "*The height of the lowest* model level is about 25 m.". Please see lines 153–154 in the revised manuscript.

6) Line 220: "and for  $O_3$  it is" instead of "and it for  $O_3$  is"

*Response:* Thanks for the constructive comment. This sentence is revised as "In China, the annual evaluation criterion for  $PM_{2.5}$  is the annual average concentration, and for  $O_3$  it is the 90th percentile of MDA8  $O_3$  concentration.". Please see lines 223–225 in the revised manuscript.

7) Line 224: "PM2.5 pollution has improved ... O3 pollution has not" or "PM2.5 pollution

improved ... O3 pollution did not"

*Response:* Thanks for the constructive comment. This sentence is corrected as "*This suggests that PM*2.5 *pollution improved significantly while O*3 *pollution did not*.". Please see lines 227–228 in the revised manuscript.

8) Line 226: "that is" instead of "that was"

*Response:* Thanks for the constructive comment. In the revised manuscript, we have corrected this typo. Please see line 230 for details.

9) Fig 3 caption: "ANAC" instead of "ANA".

*Response:* Thanks for the constructive comment. In the revised manuscript, we have corrected this typo. Please see line 261 for details.

10) Line 301: "troposphere" instead of "tropospheric atmosphere"

*Response:* We accept your suggestion and this sentence is clarified as follows: "*We first compare* vertical profiles in the model with the sounding data to determine whether the model captures the vertical structure of the **troposphere**.". Please see lines 307–308 in the revised manuscript.

11) Line 308: Remove (0.44) and (0.77) from the sentence, since these numbers are already contained in the main sentence.

*Response:* Thanks for the constructive comment. (0.44) and (0.77) have been removed. The revised sentence is "*The correlation coefficients (COR) of PM*2.5 and O3 are 0.44 and 0.77, respectively.". Please see lines 314–315 for details.

12) Line 355: I guess you mean "downward" instead of "downstream".

*Response:* Yes, it is "downward" instead of "downstream". We have corrected this typo. Please see line 364 for details.

13) Lines 465, 498, and 501: "While" is usually used to start a sub-clause, but not a main clause without a sub-clause. You probably mean something like "however" or "on the other hand", e.g., "On the other hand, anthropogenic emissions ...".

*Response:* Thanks for clarifying the usage of "while". These sentences have been clarified as "*On the other hand*, *anthropogenic emissions in Chengdu increased premature mortalities* ..." (lines 475–476), "*However*, *O*3 *pollution was likely to occur in warm months* ..." (lines 510–511) and "*O*3 *exhibited strong diurnal variation with* ..." (line 513), respectively.